# Gene Therapy in ALS and SMA: Advances, Challenges and Perspectives

**DOI:** 10.3390/ijms24021130

**Published:** 2023-01-06

**Authors:** Jan Lejman, Kinga Panuciak, Emilia Nowicka, Angelika Mastalerczyk, Katarzyna Wojciechowska, Monika Lejman

**Affiliations:** 1Student Scientific Society, Independent Laboratory of Genetic Diagnostics, Medical University of Lublin, 20-093 Lublin, Poland; 2Independent Laboratory of Genetic Diagnostics, Medical University of Lublin, 20-093 Lublin, Poland

**Keywords:** gene therapy, motor neuron disease, SMA, ALS

## Abstract

Gene therapy is defined as the administration of genetic material to modify, manipulate gene expression or alter the properties of living cells for therapeutic purposes. Recent advances and improvements in this field have led to many breakthroughs in the treatment of various diseases. As a result, there has been an increasing interest in the use of these therapies to treat motor neuron diseases (MNDs), for which many potential molecular targets have been discovered. MNDs are neurodegenerative disorders that, in their most severe forms, can lead to respiratory failure and death, for instance, spinal muscular atrophy (SMA) or amyotrophic lateral sclerosis (ALS). Despite the fact that SMA has been known for many years, it is still one of the most common genetic diseases causing infant mortality. The introduction of drugs based on ASOs—nusinersen; small molecules—risdiplam; and replacement therapy (GRT)—Zolgensma has shown a significant improvement in both event-free survival and the quality of life of patients after using these therapies in the available trial results. Although there is still no drug that would effectively alleviate the course of the disease in ALS, the experience gained from SMA gene therapy gives hope for a positive outcome of the efforts to produce an effective and safe drug. The aim of this review is to present current progress and prospects for the use of gene therapy in the treatment of both SMA and ALS.

## 1. Introduction

Gene therapy is a therapeutic tool that could ensure a lifelong therapeutic effect. It is based on delivering functional genetic material to cells in which the appropriate product will be synthesized continuously. The advantage of this therapeutic strategy over others is that it does not require multiple doses of the administered drug [1]. RNA-targeted therapies determine an important role in drug discovery. 

RNA-modifying therapy is based on antisense oligonucleotides (ASOs) and small interfering RNAs (siRNAs), and viral vectors are examples of unique methods used in the described treatment strategy [1,2]. ASOs are single-stranded deoxynucleotide analogs. They can modify gene expression by regulating protein translation, binding RNA, or interfering with the splicing mechanism. There are different classes of ASOs, including: first-class single-stranded ASOs that degrade specific RNA or modulate its metabolism by the enzyme RNaseH, and second-class double-stranded synthetic oligonucleotides which degrade RNA via an RNA-induced silencing complex (RISC) [3]. ASO-based therapies are currently used to treat a variety of conditions, for instance: mipomersen for homozygous familial hypercholesterolemia, fomivirsen for cytomegalovirus retinitis, miravirsen for hepatitis C virus (HCV), eteplirsen for Duchenne muscular dystrophy and Spinraza for spinal atrophy muscles [2,4,5]. The main limitations of ASO-based therapeutics are their short half-life, the need for lifelong repeated dosing, and their inability to cross the blood-brain barrier [1,3]. Small interfering RNAs are RNA duplexes, which, by including strands leading to the RNA-induced silencing complex, cause gene silencing, called RNA interference. Small interfering RNAs can be designed to be fully complementary to the mRNA of a target gene [2]. Small interfering RNA therapy is most commonly employed in cancer treatment, but it also has promising results in the treatment of genetic and neurological diseases [5,6]. The binding of ASOs and siRNAs to target RNAs has many similarities, and it can be used for gene silencing through various intracellular mechanisms that lead to gene silencing [4]. Viral vectors can be used in gene therapy due to their ability to deliver therapeutic genes to the central nervous system. This method uses adenoviruses, rotaviruses and the herpes simplex virus (HSV). They must present low immunogenicity and low genotoxicity and their effectiveness depends on the type of vectors. Of great importance are adeno-associated viruses (AAVs) which are promising non-pathogenic vectors that are used in the treatment of neurodegenerative diseases. AAV is single-stranded DNA belonging to the Parvoviridae family and having multiple serotypes. After systemic administration of one of them—AAV9, high expression was observed in the cervical spinal cord, hippocampus, motor cortex, cerebellum and substantia nigra, which significantly indicates the possibility of using this method in diseases of the nervous system [2,7]. SMA and ALS are some of the diseases in which the above-mentioned methods are used in the treatment of motor neuron diseases (MNDs) [1,2] (Figure 1).

Motor neuron diseases (MNDs) are a group of neurodegenerative disorders, which, among others, include amyotrophic lateral sclerosis (ALS) and spinal muscular atrophy (SMA). The two diseases differ in etiology, age of onset, survival rates and progression rates. However, both lead to a common symptom of motor neuron degeneration, resulting in muscle wasting, progressive respiratory failure and premature death. For this reason, therapies used for one of these diseases may become a starting point for the development of new forms of treatment for the other. Currently, gene therapy is a strategy with a promising prognosis for the treatment of genetically inherited and sporadic forms of both disorders [1,2]. 

This review aims to briefly present these diseases and to present the current advances and future perspectives in the use of gene therapy in the treatment of selected MNDs.

## 2. Spinal Muscular Atrophy

Spinal muscular atrophy (SMA) is an autosomal recessive neurodegenerative disease, caused by a deficiency of survival of motoneuron protein (SMN). The impairment of the production of the full-length SMN protein is the result of a mutation or deletion of the *SMN1* gene located on the 5q13.2 chromosome [8]. 

The SMN protein plays a key role in the pathogenesis of SMA. It participates in maintaining cell homeostasis. SMN is responsible for the proper formation of the spliceosome and ribonucleoprotein biosynthesis [9]. It may enable the assembly of RNA charge molecules with the various proteins required for efficient transport or local translation [10]. SMN modulates the expression of many elements that are involved in DNA repair or have anti-apoptotic effects [11]. SMN protein deficiency leads to degeneration of ⍺-motoneurons in the anterior horns of the spinal cord and, consequently, to gradual muscle atrophy [12]. In addition, in more severe forms of SMA, additional cell and tissue types are known to be affected, causing symptoms unrelated to motor neurons. The number of chondroblasts in the hypertrophic zone of the growth plate is significantly reduced, resulting in impaired bone development [13]. Defects in angiogenesis and vascular maturation, secondary to SMN deficiency, strengthen motor neuron hypoxia, thereby contributing to the pathogenesis of SMA [13,14]. SMA is a progressive and heterogeneous disease. Age of onset, clinical severity and life expectancy determine the five types of SMA. Regardless of the subtype, the disease burden of SMA is significant and affects patients in complex ways. Without appropriate therapy, it leads to muscle weakness, paralysis and, consequently, in severe cases, to death as a result of respiratory failure [15,16]. 

The complete absence of the SMN protein with the loss of both *SMN1* and *SMN2* genes is a lethal defect. More copies of *SMN2* are usually associated with higher levels of SMN and a milder course of the disease. SMN2 is a gene almost identical in sequence to SMN1. The key difference is the C-T substitution at position 6 in exon 7 (C6U), which leads to alternative splicing and exclusion of exon 7 more often (80–85%). In the process of translating the truncated *SMN2* transcript, an unstable and partially functional SMNΔ7 protein is formed (Figure 1). When the structure of the *SMN1* gene is abnormal, the full-length SMN resulting from *SMN2* translation is not able to completely prevent the onset of SMA symptoms [16].

Mechanisms involving the alternation of *SMN2* splicing by restoring the inclusion of exon 7 are targeted by gene therapy currently used in the treatment of SMA [17]. Of the positive regulators of *SMN2* splicing, the best-characterized splice factors at present are serine/arginine-rich splice factor 1 (SRSF1) and transformer beta homolog 2 (Tra2B) that bind directly to the exon 7 splicing, enhancing regions of SE1 and SE2. Heterogeneous nuclear ribonucleoprotein A1 (hnRNP A1) and the src-related substrate at mitosis 68 (Sam68) are reported to have inhibitory activity on *SMN2* splicing. Exon 7 exclusion can occur by binding the intronic N1 silencer sequence (ISS-N1) or the hnRNP A1 sequence to SE2. The C6U substitution also causes A1 or Sam68 hnRNA to bind to SE1 in place of the positive regulator [18,19,20]. Other factors involved in splicing regulation include SRp30c, TDP-43, TIA1, hnRNP Q and hnRNP G [21]. Another cause of exon 7 skipping may be the increased activity of a regulatory sequence located at the 3’ end of exon 7, called terminal stem loop 2 (TSL2). It regulates splicing in a negative way, possibly competing with U1 snRNP for the binding site [22]. Modification of *SMN2* exon 7 splicing by targeting its regulators has a proven therapeutic benefit [22,23].

At this point, it is also worth mentioning other effective scales that are successfully used in patients with SMA. The CHOP-INTEND (Children’s Hospital of Philadelphia Infant Test of Neuromuscular Disease) and HINE (Hammersmith Infant Neurological Examination) scales are used to assess the effects of treatment in patients with SMA. According to the CHOP-INTEND scale, functional assessment consists in the analysis of spontaneous or intentional movements and the observation of fine motor skills. It requires the patient to perform 16 postures. The test can be scored from 0 to 64 points (0–4 points per item). It is mainly used to assess the most severe types of SMA [24]. The HINE is a clinical neurological assessment test for infants from 2 to 24 months of age whose cutoff scores for predicting walking or sitting ability can provide important prognostic information for future motor development. The test consists of three parts: (1) neurological examination (26 items, scored on a scale from 0 to 3; total points—78) assessing cranial nerve function, posture, movements, tension, reflexes and reactions, (2) motor milestones (8 items unrated) and (3) behavior (3 items unrated). The total score can be classified as optimal (>73) or suboptimal. It also has modifications to assess the motor function of people with milder types of SMA [25]. The Motor Function Measure (MFM) is another validated tool for the numerical measurement of motor skills of patients with SMA types 2 and 3. The MFM tests both fine and gross motor skills, i.e., lifting the head, changing the position from lying to sitting, holding coins, tearing a piece of paper or drawing loops. Each task is scored from 0 to 3 points [25]. Another tool used to assess patients with SMA types 2 and 3 is the Hammersmith Extended Functional Scale (HFMSE). The assessment elements included in the HFMSE have proven to be extremely useful in clinical practice as a rehabilitation assessment tool, as well as in clinical trials to determine disease progression. The maximum number of points to be obtained is 66 [26].

### 2.1. Nusinersen

The Food and Drug Administration (FDA) and the European Medical Agency (EMA) approved nusinersen for the treatment of SMA as one of the first drugs in December 2016 and in June 2017, respectively. This drug belongs to the ASO group and leads to the inclusion of exon 7 in the *SMN2* mRNA transcripts by silencing the splicing in the SMN2 intron 7. The ISS-N1 sequence is located in exon 7, downstream of the 5’ss. This sequence inhibits exon 7 inclusion by binding to position 10–24 of intron 7. It has been shown that the use of nusinersen can increase the level of the SMN protein by blocking ISS-N1 and consequently blocking hnRNP, resulting in the incorporation of exon 7 into *SMN2* transcription. The increasing amount of functional SMN protein slows down the progression of the disease [17,27]. It is administered intrathecally in a frequency of four times over two months in the initial loading period and every four months in the maintenance period, in doses to target the central nervous system, and does not cross the blood-brain barrier if administered subcutaneously or intravenously. In order to demonstrate the effectiveness of nusinersen, a lot of research was necessary [18,23].

The first clinical trial proving the efficacy of nursinersen, the ENDEAR trial (ISIS = 396443), started in 2014. The trial included 121 infants younger than 7 months of age with two copies of *SMN2*. In an interim analysis of the Hammersmith Infant Neurological Examination (HINE) study, infants treated with nusinersen compared to controls showed a significant improvement in achieving motor milestones (41% vs. 0%, *p* < 0.001). In addition, the probability of an event-free survival, i.e., the use of assisted ventilation or time to death, was higher in the group of children treated with nusinersen (*p* = 0.005) [28,29,30,31].

The second clinical trial is the ongoing NURTURE trial (ISIS396443) which started in 2015. The study has so far included 25 infants under 6 weeks of age with genetically documented 2 and 3 copies of *SMN2*. The results of the 25-months analysis compared to the expected results from the natural course were satisfactory. During this period, all patients stayed alive and did not require constant ventilation. In addition, it has been shown that at the average age of 34.8 months, 92% of patients were able to walk with an assistant, out of which 88% were able to do so on their own. All patients survived [28,29,32].

The CHERISH clinical trial analyzed 126 children with type 2 SMA. When analyzing the results on the Hammersmith Functional Motor Scale-Expanded (HFMSE) scale after 15 months of treatment, the group of children treated with nusinersen obtained an average increase of 4 points, while the control group obtained a decrease of 1.9 points (*p* < 0.001). In the final analysis, after the same time, 57% of the children achieved a three-point increase in the HFMSE scale [25,26,27,30].

The clinical trials discussed above have shown the efficacy and safety of nusinersen. Upper respiratory tract infections are the most common complication of treatment by this drug. Other less common symptoms include headaches, backache, and atelectasis. It is recommended to test prothrombin time, platelet count and spot urine protein prior to administration due to the potential for renal toxicity and the risk of thrombocytopenia [33].

The effectiveness and safety of higher doses of nusinersen is currently being investigated in the DEVOTE clinical trial (NCT04089566). This study is divided into three parts. Part A includes patients with late-onset SMA symptoms (>6 months). Participants receive 3 × 28 mg loading doses and 2 × 28 mg maintenance doses. Part B is a randomized, double-blind, active-controlled study with infants and later-onset SMA patients. Control subjects receive 4 loading doses and 2 maintenance doses of the FDA-approved 12 mg dose. Participants in the experimental trial are given 2 loading doses of 50 mg followed by 2 maintenance doses of 28 mg. The effect of a higher dose of nusinersen on the motor functions of patients is examined by evaluating the CHOP-INTEND scale at the beginning and on day 183 of the study. Part C participants receive the approved 12 mg dose for at least one year, followed by a single 50 mg bolus dose and two 28 mg maintenance doses. This is an open-label part that studies the safety and tolerability of higher-than-approved doses of the drug. DEVOTE is followed by an open-label ONWARD trial (NCT04729907) evaluating the long-term effects of higher doses of nusinersen.

In addition, in more recent literature there is an emphasis on discovering new biological markers that can be used to assess patient progress Additionally, research suggests that miRNAs may act as primary modulators of SMN-mediated molecular pathways. Moreover, inflammatory molecules may represent new potential therapeutic targets as well as reliable biomarkers useful for patient stratification, predicting disease progression and monitoring response to therapies, and consequently better treatment of SMA patients. At this point, it is worth mentioning the study by Bonanno et al. [34], which showed that nusinersen also decreased skeletal muscle-specific miRNA levels in 21 patients with SMA types 2 and 3, associated with the pathogenic process in neuromuscular diseases. Downregulation of these miRNAs correlated with improved motor function of patients evaluated by HFMSE [34]. In another study of 21 pediatric patients with SMA types 1, 2, and 3; and 12 adults with SMA types 2 and 3, Bonanno et al. [35] showed that nusinersen therapy may have a beneficial effect on the peripheral immune system. After 6 months of treatment, a decrease in the level of pro-inflammatory cytokines of the Th1/Th17 pathway was noted in the serum of patients. Interestingly, the above studies identified miR-133a and IL-23 molecules as potential predictive biomarkers of nusinersen therapy, and IL-10 as a potential biomarker for on-treatment monitoring [34,35].

### 2.2. Gene Replacement Therapy

The second drug approved for SMA therapy is Zolgensma (AVXS-101, Onasemnogene Abeparvovec). In 2019, the FDA authorized the use of Zolgensma for children under 2 years of age whose diagnosis of SMA has been confirmed by genetic tests. It is gene replacement therapy (GRT), which was created due to knowledge of the genetic basis of the disease [36]. It uses a non-replicating adeno-associated virus capsid (scAAV9) to deliver transgene under a ubiquitous promoter to motor neuron cells. It is noteworthy that, in contrast to nusinersen, Zolgensma crosses the blood-brain barrier and one administration per 1 h intravenous infusion is sufficient for the systemic expression of the SMN protein [28].

Adeno-associated viruses (AAV) are non-enveloped viruses derived from the genus Parvoviridae. Their nucleic acid is in the form of single-stranded DNA [37,38], and they have two open reading frames in their genome, which are the cap and rep genes, flanked by inverted terminal repeats at both ends. The cap gene produces VP1, VP2 and VP3, which are the structural proteins that form the capsid. In contrast, the rep gene encodes proteins involved in AAV replication, transcription, integration and encapsidation [37]. However, rep and cap genes are not yet included in any AAV-based therapeutics.

To date, more than 12 different AAV serotypes have been isolated. They vary in cell tropism (depending on the type of capsid surface proteins), differences in transduction efficiency and differences in immune response capacity. All AAVs are capable of transducing distinguished dividing and non-dividing cells and combine low immunogenicity and lower pathogenicity with long-term transgene expression in clinical applications. What is more, many AAVs successfully transduce neurons and glial cells, which enabled the use of vectors derived from them in neurodegenerative diseases [8,39].

In this context, we can distinguish the AAV2 serotype, which is specific for cerebral endothelial cells, or the AAV9 serotype, which after systemic administration leads to high expression in neurons of the motor cortex, cerebellum, substantia nigra and cervical spinal cord [7,8,40].

However, they also have some limitations. One of them is the fact that AAVs are prone to reducing their effectiveness through the influence of the produced neutralizing antibodies. Moreover, receptors of various AAV serotypes are found in many organs, which limits their target specificity [8]. Nevertheless, this gene expression specificity of AAV delivery can be achieved by using an organ-specific promoter [41,42]. It is worth mentioning that in comparison to LV vectors, AAVs are characterized by a higher safety profile and the level of expression of transgenes. Additionally, AAVs have greater vector spread and better stability of their genome mainly as extrachromosomal episomes, which reduces the possibility of insertional mutagenesis [43,44].

In the first phase of clinical trials in the START trial, the therapy was administered to 15 patients with SMA type 1. Participants were divided into two groups. One cohort (12 people) received the higher dose −2.0 × 10^14^ vg/kg and the other cohort (3 people) received the lower dose −6.7 × 10^13^ vg/kg (NCT02122952). The mean age of patients in the higher dose cohort was 3.4 months and in the lower dose cohort was 6.3 months. All patients in both groups lived to at least 20 months of age and did not require permanent mechanical ventilation. This proved to be a significant difference compared to the historical cohort in which only 8% of patients survived the threshold of 20 months of life without constant ventilation. Subjects in relation to historical cohorts had a longer event-free period, achieved milestones faster, and their motor function improved. The patients treated with the higher dose scores in the Children’s Hospital of Philadelphia Infant Test of Neuromuscular Disorders (CHOP-INTEND) increased by an average of 15.4 points over three months. A total of 92% of the higher dose cohort were able to sit without assistance, 92% could take food orally and speak, 75% rolled over and 17% walked independently [45]. A later analysis showed that children who received early GRT had a higher increase in CHOP-INTEND than older patients at the time of Zolgensma administration [46].

Phase three study—STR1VE tested the safety and efficacy of Zolgensma on patients with *SMN1* mutation with one or two copies of *SMN2* from different centers around the world (NCT03306277). Results from 22 patients in the US (STR1VE-US) compared with studies from cohorts with a natural history of the disease confirmed the efficacy of GRT. 91% of patients at 14 months did not require permanent ventilatory support and 59% of patients at 18 months sat unassisted for 30 s or more. It has also been proved that the therapeutic benefits of Zolgensma treatment outweigh the risks associated with possible side effects [47].

Clinical trials for intrathecal administration of Zolgensma are currently underway to investigate long-term safety and efficacy or to explore therapeutic potential using other routes of administration [NCT05089656]. In pre-clinical studies, intrathecal injection of AAV9 carrying human SMN cDNA into mice and non-human primates resulted in similar targeting of motor neurons using a 10-fold lower dose compared to intravenous injection [48].

The serious side effects that may occur after administration are consistent with those observed with other AAV9 based therapies. An increase in liver enzymes and vomiting has been reported, which was reversible with adequate doses of prednisone or prednisolone [49]. Elevated troponin I levels and thrombocytopenia were also observed after Zolgensma treatment. There have also been reports of thrombotic microangiopathy associated with the Zolgensma [50,51]. The above observations emphasize the importance of monitoring liver and kidney function at least 3 months after GRT administration [45].

The findings in mouse models reveal that long-term AAV9-SMN GRT may have the opposite effects, inducing late neurotoxicity in the same neurons where it provides early functional correction of SMA deficits, through late motor dysfunction associated with loss of proprioceptive synapses and neurodegeneration. This also shows that both a severe excess and deficiency of SMN have deleterious effects on sensorimotor circuits. With the probable irreversibility of persistent SMN overload in neurons caused by AAV9-SMN GRT, awareness of unexpected therapy scenarios will result in a deeper evaluation of the costs and benefits of gene therapy compared to other available options [50].

### 2.3. Risdiplam

Risdiplam (Evrysdi™) is an orally administered *SMN2*-directed splicing modifier. It was approved by the FDA on the 7th of August 2020 to treat patients with the *SMN1* mutation who were 2 months old or older [52]. At present, it is approved for the treatment of SMA in both children and adults of all ages [53]. Risdiplam is a small molecule binding directly the exonic splicing enhancer 2 (ESE2) to exon 7 and the 5 ‘splicing site (5’ss) of intron 7 in the *SMN2* transcript. This displaces the hnRNPG, facilitating the binding of U1 snRNP to 5’ss. In this way, risdiplam promotes the inclusion of exon 7, thereby increasing the level of FL-SMN [54]. In in vitro and in vivo studies, risdiplam also affected the *FOXM1* and *MADD* genes, which may influence cell cycle regulation, apoptosis and cause adverse effects in animal models [52,55]. Risdiplam also promotes the activation of *SMN2* splicing by binding to the splicing modulators of the pre-mRNA *SMN2* complex—far upstream element binding protein 1 (FUBP1) and KH-type splicing regulatory protein (KHSRP) [56]. The advantage of risdiplam is that it penetrates the blood-brain barrier, increasing the level of SMN protein in both the central nervous system and peripheral organs [57]. This small molecule is administered orally at the same time or after a meal. The dose depends on body weight, and the age of the patient may affect the pharmacokinetics of the drug [52].

The first part of the open-label, multi-center clinical study FIREFISH (NCT02913482) was designed to assess the safety, tolerability, pharmacokinetics and pharmacodynamics of different doses of risdiplam. This clinical trial enrolled 21 infants with SMA type 1, four of whom received a low dose (final dose at month 12 is 0.08 mg of risdiplam per kilogram body weight per day) and 17 received a high dose (final dose at month 12 is 0, 2 mg per kilogram per day). A total of 19 infants were treated for a minimum of 12 months and the mean duration of treatment was 14.8 months. Higher blood levels of SMN were observed in the cohort treated with the higher dose. Seven infants in the high-dose cohort and no infants in the low-dose cohort were able to sit unassisted for at least 5 s [58]. In the natural history of the infantile-onset SMA, the mean survival age is 8 months [59]. At 16 months, 14 patients in the higher dose cohort had a CHOP-INTEND score of 40 or higher, and two infants achieved HINE-2 walking assessment milestones of bouncing [58]. The second part of the study confirmed the efficacy and safety of the higher dose. The study comprised 41 patients, 38 of them were alive at month 12, and three required assisted ventilation. The ability to sit unassisted was achieved by 12 children [60].

The SUNFISH study (NCT02908685) included patients with a later onset of symptoms. Risdiplam has been given to people between 2 and 25 years of age with type 2 or 3 SMA. The first part of the study was to increase the dose. After 24 months, motor function significantly improved, resulting in a higher Motor Function Measure (MFM) score than the comparable cohort [61]. In the second part, the effect of risdiplam’s dose selected in the first part was compared to placebo. After 12 months of therapy, compared to the placebo group, patients who received risdiplam achieved much greater baseline improvement in Motor Function Measure 32 (MFM32), HFMSE, SMA Independence Scale (SMAIS) and Revised Upper Limb Module (RULM) [62].

JEWELFISH (NCT03032172) was a multicenter, open-label study that tested the pharmacokinetics, pharmacodynamics, efficacy and safety of risdiplam in a group of subjects with a confirmed diagnosis of 5q-autosomal recessive SMA. Patients in the study were 6 months to 60 years of age and had previously received RG7800 (RO6885247), nusinersen, olesoxime, or Zolgensma. After 12 months of therapy, an increased level of SMN in the blood of patients was observed [63].

An ongoing trial, RAINBOWFISH (NCT03779334), recruits pre-symptomatic infants with genetically confirmed SMA from birth to 42 months of age. The study aims at evaluating efficacy, safety, pharmacodynamics and pharmacokinetics. The primary endpoint is the proportion of seated infants after 12 months of treatment, and secondary endpoints include long-term assessments of motor performance and other developmental milestones [64]. This study will provide valuable information on the effects of treatment and the selection of the optimal method of conducting risdiplam therapy in the pre-symptomatic phase in small infants with SMA and will support decision-makers in the importance and value of pre-symptomatic therapy in the case of SMA [64] (Table 1).

The most common adverse reactions that occurred during clinical trials were fever, diarrhea, rash, constipation, nausea, vomiting, headache, mouth ulcers and sores, joint pain or urinary tract infections [55,65,66].

### 2.4. Challenges and Future Directions

The other orally administered small molecules tested in SMA patients were RG7800 (RO6885247) and branaplam (LMI070, NVS-SM1). Similar to nusinersen and risdiplam, they elevate SMN levels through the alternation of *SMN2* splicing. The RG7800 phase 1 clinical trial was terminated because the patients developed retinotoxicity [66]. During the trials, branaplam showed toxic effects on the nervous system, kidneys and vessels [67]. RG7800 and branaplam clinical trials participants were enrolled in the JEWELFISH study.

An interesting approach also seems to be the one presented by d’Ydewalle et al., whose efforts focused on an attempt to increase the expression of SMN. As an intermediary for this, they used a long non-coding RNA (lncRNA) derived from the antisense strand of the SMN (SMN-AS1), which represses SMN expression. By degrading SMN-AS1 with ASOs, they increased SMN expression in patient-derived cells, cultured neurons and the central nervous system of mice. As a result, it improved the survival of mice with severe SMN and may successfully become the basis for further research on the use of combinatorial ASOs in the treatment of neurogenetic disorders [68].

Antisense phosphorodiamidate morpholino oligomers (PMO) are also tested. PMO is more stable in vivo and is present in muscles longer than ASOs. Studies in a severe SMA mouse model have shown that PMO injected into the lateral ventricle promotes exon 7 inclusion [69].

Delta neurocalcin (NCALD) belongs to the calcineurin family and negatively regulates endocytosis. The protective effect of NCALD downregulation on the SMA phenotype has been confirmed in animal models of SMA. The combination treatment of subcutaneous injection of SMN-ASO and intraventricular injection of NCALD-ASO has recently been shown to improve motor dysfunction in SMA mice [70]. PLS3 contains two interacting potentials: 1C, the overexpression of which amplifies the axonal coronin in the zebrafish model of SMA, and the calcineurin B homolog 1, which further supports downregulation endocytosis and expansion of SMA in mice [71].

Other targets for gene therapy may be the ZPR1 (ZPR1 Zinc Finger) gene and the PLS3 (Plastin 3). Tests in SMA mice models suggest that overexpression of PLS3 or ZPR1 protein may ameliorate the course of the disease [72,73].

## 3. Amyotrophic Lateral Sclerosis

Amyotrophic lateral sclerosis (ALS) is one of the most common MNDs. According to Mehta et al., the incidence of this disease in the USA in 2014 was 5 per 100,000 people and did not change compared to 2013 [74]. Additionally, the meta-analysis presented by Marin et al. defined the worldwide standardized incidence of ALS as 1.68 per 100,000 person-years of follow-up [75]. Treatment prognosis is not promising—the survival rate is 2 to 5 years. Only 5% of patients survive 20 years from the time of diagnosis. The condition manifests by progressive loss of function of the motor neurons and glial cells in the spinal cord and in the brain. Paralysis appears as a consequence of the progress of ALS. Neuromuscular weakness affects men more often, usually after the age of 50 [76,77,78]. In the basic classification, we can determine the family form (fALS), which occurs in 5–10% of cases and the sporadic form (sALS), which includes all other patients. More advanced classification includes: (i) primary lateral sclerosis with upper motor neuron involvement (ii) limb-onset ALS with a combination of low motor neuron and upper motor neuron (iii) pure involvement of low motor neuron-progressive muscular atrophy (iv) bulbar-onset ALS [76].

Another classification is used for phenotypic differentiation. It consists of distinguishing groups depending on the part of the body where symptoms appear. Most patients report symptoms starting in the extremities—fasciculations, cramps. Some of them report dysphagia as a bulbar symptom, while only 5% show symptoms of the respiratory muscles or the trunk. Respiratory failure affects, among others, the diaphragm, which is manifested by shortness of breath and hyperventilation. Most patients die of respiratory failure [78]. Despite many decades of clinical trials, there is no effective cure. However, drugs that are currently being used allow to marginally increase survival time. Some of them are riluzole and edaravone [79]. The first, riluzole, was approved for the treatment of ALS more than 20 years ago. Its action is based on an inhibitory effect on glutamate release, which in turn leads to the inactivation of voltage-gated sodium channels and disruption of the downstream steps following neurotransmitter binding. Although its exact mechanism of action in ALS is unknown, many studies have shown its significant effect in delaying the time to tracheostomy and prolonging survival by preventing the destruction of nerve cells [80,81,82]. In turn, edaravone, by acting as an antioxidant and free radical scavenger, reduces excessive oxidative stress and inhibits cell death. Thus, it prolongs function and delays motor deterioration, especially in patients in the early stages of ALS [81,83,84]. It is worth mentioning that in May 2022, the FDA also approved an oral formulation of edaravone for use in patients with ALS, which is undoubtedly much more convenient than intravenous dosing of the drug. In addition, a global phase 3 study evaluating the safety of edaravone in a group of 185 patients has recently been completed. In the results, after 48 weeks, treatment-related adverse events of falls, muscle weakness and constipation were observed. Although serious events (such as dysphagia, respiratory failure) were reported by 25.9% of patients, no serious adverse events were related to the study drug, confirming that edaravone is well-tolerated during 48 weeks of treatment [83,85]. At the core of ALS are complex molecular mechanisms that may be a therapeutic target for treatment. Various pathogenic variants have been detected in genes encoding RNA binding proteins (RBPs), e.g., *TAF15*, *ATXN2*, *FUS* and *TDP-43* in the DNA of the patients [86]. Gene therapy with riluzole and edaravone, the only two known agents shown to slow down disease progression in ALS, for two forms of the disease—one caused by mutations in the C9orf72 gene and the form related to superoxide dismutase 1 (SOD1) Cu/Zn—may be a promising treatment therapy in the case of failure of pharmacological approaches.

The role of the *SOD1* gene is based on protecting the cell against reactive oxygen species and it is also the first and most studied gene associated with ALS. The protein derived from it exists as a homodimer, in which each monomer is characterized by a high structure with disulphide bridges increasing their stability. However, the structure of the protein may become destabilized. This is due to mutations in the *SOD1* gene, which may lead to the breakdown of the protein’s homodimeric structure, thereby enhancing the toxic function of SOD1 proteins. It has been shown that 12% of patients diagnosed with the familial form have mutations in the *SOD1* gene, and it can occasionally occur in sALS as well. Although the potential mechanisms of the toxicity of mutated *SOD1* in motor neurons are still not fully understood, the *SOD1* gene itself marked the beginning of a new era of research on ALS. This is due to the fact that it was used to create the first SOD1-G93A transgenic mice model, which is still used in clinical trials [79,87,88,89].

The second mentioned mutation is based on the hexanucleotide repeat (HRE) expansion of GGGGCC (G4C2) in the first intron of the *C9orf72* gene. It can be detected in approximately 40% of all fALS and in a large number of seemingly sporadic cases, making it the most common single mutation diagnosed in this disease. There are a number of possible mechanisms by which this mutation may contribute to the development of ALS. In this context, we can distinguish: the down-regulation of *C9orf72* gene expression leading to loss of function; movement of repeat containing RNA into the cytoplasm where it can be translated into dipeptide repeat proteins (DPRs); and bi-directional transcription of HRE into RNA containing the G4C2 and C4G2 repeats that aggregate in the nucleus of cells to sequester RNA binding proteins Figure 2 [90]. Although the C9orf72 protein has been shown to be highly expressed in neurons, it is still unclear which of these mechanisms is the most important in the pathogenesis of ALS. Despite this, modulation of the *C9orf72* gene plays a significant role in ALS research using ASO and genome editing [79,91,92].

### 3.1. RNA Interference

The identified ALS-related mutations may cause dysregulation in mRNA processing [93]. There have been several trials of a therapeutic approach for ALS using siRNA. A major challenge for research is to optimize the delivery and biodistribution of nucleic acids in vivo. One of the first studies demonstrated that siRNA delivered to spinal motor neurons by AAV-2-mediated retrograde transport from the muscles had the effect of reducing SOD1 levels in the ALS mouse model [94]. Further studies on the effect of siRNA targeting *SOD1*, transported by various viral vectors, resulted in delaying the onset of disease symptoms and prolonging survival of the tested mice [95,96,97,98]. In another attempt, Rizvanov et al. [99] administered siRNA locally to the sciatic nerve of the mouse model. Treatment with specific siRNA resulted in a 48% decrease in the level of human *SOD1* mRNA in the lumbar region of the spinal cord [99]. Non-human primates given rAAVrh10-miR-SOD1 were also studied. Significant and safe silencing of *SOD1* has been achieved [100]. In 2017, adenovirus associated (AAV10) related viral vectors of rh10 serotype (AAV10) were used to mediate hSOD1 pre-mRNA exon skipping by expressing exon-2 targeting antisense sequences embedded in modified U7 micronucleus RNA (AAV10-U7-hSOD). A decrease in SOD1 levels was obtained by alternative splicing, omitting the *SOD1* exon 2. AAV10-U7-hSOD was delivered intravenously and intraventricularly. The effects observed in mice, such as preventing weight loss and impaired neuromuscular function, as well as increased survival, support the effectiveness of this technology [101]. The therapeutic approach using RNAi has the potential to be another form of gene therapy for ALS but still needs to overcome such difficulties as in vivo instability, lack of siRNA specificity and potential toxicity of RNAi-based technologies [102].

### 3.2. Antisense Oligonucleotides

Lessons learned from SMA therapy have led to the use of selective antisense oligonucleotides (ASOs) targeting RNA in therapeutic approaches to treat familial types of ALS. These short, single-stranded nucleic acids can affect the gene by modifying splicing or inducing mRNA degradation by activating ribonuclease H (RNase H).

In 2006, Smith et al. [103] gave rats an intraventricular ASO targeting SOD1. The study showed a good distribution of ASO within the central nervous system, effective suppression of rat *SOD1* mRNA, and decreased levels of SOD1 protein in nervous tissues. It was also associated with slowing the course of the disease and prolonged survival [103]. Phase 1 study confirmed the safety and good tolerability of intrathecal ASOs in humans. The conservative doses (3 mg) used in the study did not alleviate disease symptoms [104,105]. Phase 1–2 studies on Tofersen (BIIB067) showed promising results. It is an ASO that reduces SOD1 protein synthesis by mediating RNase H dependent *SOD1* mRNA degradation. Clinical trials have shown that higher doses are effective in reducing central nervous system SOD1 levels. Exploratory studies may suggest that a 100 mg dose of Tofersen slows the decline in the Amyotrophic Lateral Sclerosis Functional Rating Scale-Revised (ALSFRS-R) [106]. For this reason, it has been suggested that a 100 mg dose of Tofersen may slow down the decline in Amyotrophic Lateral Sclerosis Functional Rating Scale-Revised (ALSFRS-R), leading to a Phase 3 study. Within 28 weeks, the drug used reduced the concentration of SOD1 in the cerebrospinal fluid and light chains of neurofilaments in plasma compared to placebo. However, it did not improve clinical endpoints (change in ALSFRS-R score) and was associated with adverse events such as myelitis [107]. Currently, the potential effects of earlier versus delayed Tofersen injection are being evaluated in an extension phase funded by Biogen [108,109].

ASOs that are able to abolish pathology and reverse toxicity mediated by *C9orf72* RNA expansion have also been identified. Treatment of ASO to reduce the number of RNAs containing HRE appears to be a rational and promising approach in the treatment of *C9orf72*-associated ALS [110,111,112]. In a study by Jiang et al. [113] a single injection of ASOs was given to BAS transgenic mice, which selectively bind to sense expansions of RNA. Decreases in sense RNA and sense DPRs and improvement in behavioral abnormalities have been observed [113]. The results of these studies ultimately led to the Phase I clinical trial of ASO BIIB078 for C9orf72-ALS adult patients (NCT03626012). Stereopure oligonucleotides have recently been designed to target a sequence called SS1b at the exon 1b—intron 1 junction of *C9orf72*. They function by selectively degrading the sense expansion of G4C2 containing the transcripts without reducing the variant 2 isoform lacking the repeat. It is also worth adding that they prevent glutamate toxicity. This type of oligonucleotide has the potential to be addressed to *C9orf72*-associated ALS [114,115,116].

The p.P525L mutation in the *FUS* gene is usually associated with the early onset and aggressive course of ALS. In 2019, a 26-year-old patient with this mutation was given a personalized ASO. The first injection was given when the patient experienced breathing difficulties due to the disease and died less than a year after starting therapy [104]. Jacifusen (ION363) is an ASO that has now entered Phase 3 clinical trials and has the potential to be a unique therapy for *FUS*-ALS patients. There are 77 patients who are given Jacifusen or placebo (NCT04768972) [117,118,119].

Targeting ASOs to *ATXN2* also appears to be a promising therapeutic strategy. The delivery of ataxin-2 targeting ASOs to neuron differentiated iPSCs from *C9orf72*-ALS patients reversed the cytoplasmic abnormal localization of nuclear proteins. Administration of *ATXN2*-ASO to the lateral ventricle after birth of the rapidly progressing murine *TDP-43* ALS model resulted in a sustained, marked reduction in *ATXN2* mRNA as well as prolonged survival and improved gait [120]. Orally administered BIIB100 is currently under clinical trials for the treatment of adults with ALS (NCT03945279) Table 2.

### 3.3. Viral Vectors in ALS

So far, gene delivery to the central nervous system has been limited for most therapeutic molecules due to the existence of the blood-brain barrier. However, even this barrier can be overcome, as demonstrated by Duque et al. through the use of AAV9 in animal studies [111]. Currently, the most commonly used are lentivirus-derived vectors and adeno-associated virus (AAV) vectors [38,79,121,122].

#### 3.3.1. Lentiviral Vectors

Lentiviruses come from the Retroviridae family. Among them, there are two groups: the primate-like, where the vector is based on the human immunodeficiency virus (HIV), and non-primate vectors, for example from equine infectious anemia virus (EIAV) [123].

Depending on the packaging plasmid used for the production of lentiviral vectors (LV), several generations can be distinguished. The last, third generation, is characterized by a simplified genome organized into the gag, env and pol genes that encode the viral envelope, viral capsid structural proteins, and proteins necessary for viral DNA synthesis. Additionally, two long terminal repeats are present, the elements which determine reverse transcription, integration into the host genetic material and gene expression [117]. However, one of the major features of the use of these viruses as gene therapy vectors is undoubtedly the elimination of many regulatory and accessory genes within them. Among them are, for example, tat and vpr, which are responsible for oncogenesis and apoptosis, respectively [124].

Compared to other vectors, LVs are distinguished by their ability to transduce dividing and non-dividing cells (including neurons), the ability to naturally penetrate the intact nuclear membrane and high cloning ability (8–10 kb) [79,102,123,125,126]. Additionally, by integrating with host cell chromosomes, LV can induce long-term and stable expression without inflammation and its tropism can be altered by genetic manipulation of glycoproteins necessary for vector transduction [79,126].

The most commonly used is the subtype with vesicular stomatitis virus (VSV-G) glycoprotein in its envelope, which is characterized by high transduction efficiency and stability [60,100]. The potential for its use was tested by Raoul et al. By bilateral spinal injection of VSV-G LV causing SOD1 silencing mediated by RNA interference, in SOD1-G93A transgenic mice of the family ALS (fALS) model, they achieved a delay in disease onset and progression [96].

Despite their many advantages, LVs are not free from disadvantages that limit their use. They are a limited area of transduction (approximately 500–700 µm from the injection site), large size (diameter 100 nm) and low viral titers [127]. However, their integration into the genome makes them promising for ex vivo gene therapy, as demonstrated by Suzuki et al. in their SOD1 rat studies [128].

#### 3.3.2. Adeno-Associated Virus Vectors

As part of ALS gene therapy, serotype AAV9 seems promising. Recent studies using its vector allowed the delivery of shRNA to SOD1 in SOD1-G93A mice, which translated into an increase in their median life [97]. In addition, the synthesis of the mutant SOD1 was reduced, which slowed the disease progression [129]. This approach was tested in preclinical studies but was put on hold due to a change in the company’s priorities [129,130,131]. However, this does not exclude the effectiveness of the chosen method of therapy, the confirmation of which requires further research.

Noteworthy is also the AAVrh10 serotype, which, due to its transduction into the central nervous system, can effectively deliver microRNA there, which, by binding SOD1 mRNA, would thus reduce the production of mutant proteins in patients with this form of ALS. Therefore, it would improve the survival and function of motor neurons, resulting in therapeutic benefits for people with SOD1-associated ALS. This was demonstrated by Wang et al. who, through its use, extended the survival of mice with the SOD1 mutation [89,130]. Similar effects were presented by Borel et al. [100] In their study, they obtained an extension of the median life expectancy of SOD1-G93A mice by 21% [100].

It is also worth mentioning the study of Biferi et al., whose molecular strategy, based on inducing a permanent reduction in the level of mutant SOD1 in SOD1-G93A mice, led to therapeutic effects. To achieve this, they used AAVrh10 vectors to mediate hSOD1 pre-mRNA exon skipping. This led to the production of a premature termination codon, and as a result, increased survival in the tested mice and prevented the deterioration of neuromuscular function in them [101].

Currently, the safety, tolerability and efficacy of AAVrh10 intrathecal administration of anti-SOD1 microRNA to patients with SOD1 ALS mutations is being tested in the multi-center clinical trial APB-102 [132,133].

In addition to animal studies, a single intrathecal AAV infusion encoding a microRNA targeting SOD1 has recently been used in two patients with fALS.

In Patient 1, this dose decreased the levels of SOD1 in the spinal cord compared to the control group but had no effect on the levels of SOD1 in the cerebrospinal fluid. Additionally, it was not possible to clearly interpret the behavior of motor neurons on both sides of the lumbar-sacral spinal cord. Although this patient regained minimal left-hand extension, other clinical features typical of ALS deteriorated. Therefore, it could not be concluded whether SOD1 inhibition played a role in improving the clinical course of the disease. In contrast, Patient 2 therapy with AAV vectors did not bring any clinical benefit. In this study, the only theoretical advantage was that gene suppression mediated by viral vectors allows for a sustained effect of a single dose of therapy. However, it is associated with the possibility of side effects. Although this study did not show satisfactory results, it did not destroy the role of AAV vectors in ALS gene therapy in humans [134].

An alternative approach could be the use of AAV vectors for the therapeutic delivery of the human gene DOK7. This is due to the fact that its expression causes the activation of muscle-specific kinase MuSK, which in turn inhibits the degradation of motor nerve endings at neuromuscular junctions. This was demonstrated by Miyoshi et al., who achieved inhibition of muscle atrophy in the SOD1-G93A ALS mice model by using an AAV vector encoding the DOK7 gene. This translated into an improvement in motor activity and an extension of the life of the tested mice [135].

However, neurotrophic factors with neuroprotective and neuro-regenerative properties may also translate into ALS patients [136,137].

For this reason, Gross et al. [137] conducted a study using recombinant AAV serotype-2-neurturin (AAV2-NRTN) to evaluate its safety, tolerability and efficacy in the SOD1-G93A ALS mice model. Following the injection of AAV2-NRTN into the cervical spinal cord, NRTN expression and thus a neuroprotective effect was observed in cervical motor neurons and at neuromuscular junctions. This was reflected in the slowing of the decline in forelimb grip strength in the tested mice. As no increase in morbidity was observed, this result encourages further continuation of research in this direction [138].

Moreover, human insulin-like growth factor 1 (hIGF1) could be used as a neurotrophic factor for therapeutic purposes in ALS. This was checked by Lin et al., who injected intramuscularly self-complementary AAV9 encoding hIGF1 into the hSOD1-G93A ALS mice model. They observed that this significantly reduced the loss of the anterior lumbar spinal cord motor neurons and delayed muscle wasting in the tested mice. It is worth noting that, in this study, IGF1 delayed the onset of the disease and prolonged the lifespan of ALS mice. Undoubtedly, this experiment supports further research [139].

As can already be seen, the use of viral vectors in ALS gene therapy is rather extensive. While promising results have been achieved in many animal studies with these vectors, many challenges still remain to be overcome. We believe that constantly improved serotypes of the vectors created will soon allow their introduction to the clinical therapy of patients with ALS.

### 3.4. Genome Editing

An approach that has recently been tested for the treatment of both genetic and non-genetic disorders is genome editing [140,141]. Its techniques include zinc finger nuclease, transcription activator-like effector nuclease and Cas9 nuclease associated with clustered regularly interspaced short palindromic repeats (CRISPR) [141]. In the context of ALS, the last one seems to be the most promising, as its functionality is increasingly being tested in clinical trials.

In order to use CRISPR-Cas, a small guide RNA (sgRNA) is first constructed close to the sequence of a proto-spacer adjacent motif (PAM). As a result, in the next step, Cas nuclease is able to bind to the target DNA sequence, which is recognized by the sgRNA. In effect, this pattern leads to a double strand break (DSB) in the cell’s DNA, inducting base insertions or deletions to cause a frameshift. These breaks are in turn joined by two different repair processes: non-homologous end joining (NHEJ) and homologous recombination (HR). The first of these techniques, NHEJ, is a fast but error-prone process that often leads to insertions and deletions at the breakpoint, thereby knocking out the gene encoding the protein. In contrast, HR induces insertions or replaces DNA fragments with a donor template, while being relatively error-free Figure 3 [142,143,144,145,146].

Depending on the Cas protein subtype, we can distinguish several types, including saCas9, c2c2, RCas, dCas or Cas9 from Streptococcus pyogenes (SpCas9) [147]. Each of these proteins has a unique PAM sequence and a different size, adapted to all kinds of applications. It is also worth adding that these proteins have specific targets for cutting DNA and RNA, and, therefore, they can induce single-stranded cuts, leading to the activation or inhibition of transcription [147].

In recent years, a lot of research has been carried out focusing on the use of genome editing in the context of motoneuron diseases. One of them was the study by Gaj et al. [148], which presented the possibility of disrupting the expression of mutant SOD1 in the ALS G93A-SOD1 mice model. For this experiment, the authors used CRISPR-Cas9, which was delivered in vivo using an AAV vector. They found that the applied genome editing reduced the mutant SOD1 protein >2.5-fold in the thoracic and lumbar spine. As a result, it improved motor function and reduced muscle atrophy in the mice tested. It is worth adding that, in the final stage, the mice tested showed about 50% more motor neurons, and about 37% of the disease was delayed [148].

Similar effects were achieved by Duan et al., who applied mutant SOD1 modification in G93A-SOD1 transgenic mice. By using the AAV-SaCas9-sgRNA system, they achieved the deletion of the SOD1 gene and the improvement in life expectancy of the mice tested by 54.6% [149].

Recently, Deng et al., demonstrated that CRISPR-Cas9 mediated genome editing prevents disease progression in SOD1 mice. Moreover, they saw no evidence of other diseases beyond the age of two in genome-edited mice. This is further proof of the efficacy of this promising therapeutic approach like CRISPR-Cas9 [150].

In turn, in the study by Pribadi et al., the CRISPR-Cas9 system was used to completely remove the large repeat expansion mutation within C9orf72 in induced patient-derived pluripotent stem cells (iPSC), which have the ability to differentiate into other types of cells, such as neurons. This action prevented RNA foci formation and promoter hypermethylation, two phenotypes of the C9orf72 mutation. However, this did not significantly alter the expression of C9orf72 at the mRNA or protein level. Nevertheless, this work opens up further possibilities for the use of CRISPR-Cas9 in cutting out the mutant C9orf72 repeat expression in ALS therapy [143,151].

A similar study based on C9orf72 was conducted by Lopez-Gonzalez et al. In their model, they presented evidence of a significantly higher expression of Ku80 (which is a DNA repair protein) in the neurons of patients derived from C9orf72 iPSC, which also increased other pro-apoptotic proteins such as Bax and PUMA. To inhibit their expression, they used the deletion of G4CA extended repeats via CRISPR-Cas9. As a result, their experiment proved successful because it inhibited the over-activated DNA repair pathway [152].

Despite the hopes placed on the possibilities of developing the CRISPR-Cas technique, it is necessary to bear in mind all their limitations, which still pose a challenge for researchers.

One of these is the fact that the system’s guide RNA does not need to be fully compatible with the target sequence for its cleavage. In practice, this means the possibility of non-targeted effects, which could translate into the occurrence of unpredictable mutations [153]. Much effort has already been made to overcome this limitation. The possible non-target effects can be reduced at the beginning of the study by selecting the right target and designing the sgRNA appropriately. In this context, the content of guanine-cytosine in the sgRNA is important, as it has been found to have a significant impact on the incidence of extra-target effects [154]. Additionally, it is possible to use a truncated targeting RNA, which is an effective and feasible possibility to increase the efficacy of Cas9 nuclease because it recognizes shorter target sequences without sacrificing effectiveness [155].

Another problem with the use of genome editing is the delivery of sgRNA and Cas9. For this purpose, mostly LV and AAV viral vectors are used, which, despite their effectiveness, exhibit limited tissue tropism and cargo capacity. Additionally, although vectors are designed in a way that should make replication difficult and avoid virulence, undesirable deleterious effects may occur. In this matter, the solution may be non-viral vectors [155].

Undoubtedly, the CRISPR-Cas technology is one of the most promising in the context of the future clinical therapy of ALS. Despite the fact that, at the present stage, it is rather limited to generating small animal models, the results of the conducted research led to further work in this direction. It is possible that, in the next few years, this method will be even better developed, which will not only reduce the expression of mutant proteins in ALS patients but also extend and improve their lives.

The results of the above-mentioned studies are collected and summarized in Table 3.

## 4. Conclusions

For many years, MNDs were considered incurable. Effective treatment has become possible thanks to gene therapy. Targeting the genetic basis via ASOs, GRT or small molecules has proven to be crucial in the treatment of SMA. Good results in patients after administration of the currently approved nusinersen, Zolgensma or risdiplam give an optimistic perspective for further studies. It was found to be of particular importance in the pathology of the same group—ALS, in which several drugs are currently undergoing clinical trials. Targeting the most common mutations related to ALS gives positive results. Research is being carried out to find a substance that would treat both the sporadic and familial forms of the disease. Currently, research efforts are focused on finding new therapeutic technologies, such as genome editing, and on testing new RNA targets for gene therapy that could optimize treatment. Attempts to implement the therapy of neurotrophic factors are also underway. Constant efforts and progress in research on the basis of MNDs and RNA-targeting molecules allow us to believe in a significant improvement in the quality of life of patients suffering from neurodegenerative diseases.

## Figures and Tables

**Figure 1 ijms-24-01130-f001:**
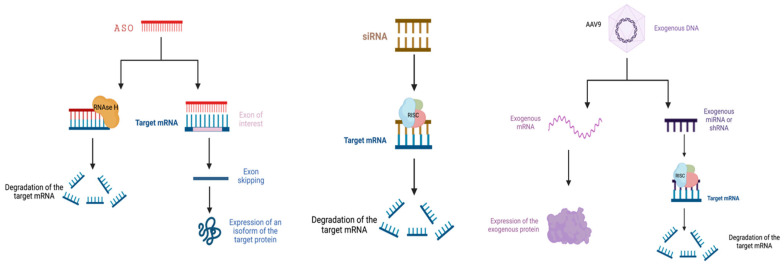
Different strategies used in gene therapy.

**Figure 2 ijms-24-01130-f002:**
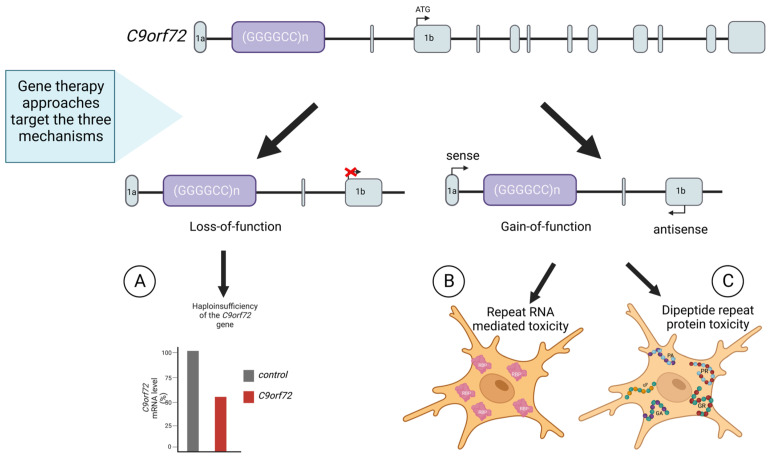
Pathomechanisms of *C9orf72*-associated ALS. (**A**) HRE can inhibit the transcription of C9orf72, resulting in loss of function. (**B**) Expansion can also be transcribed bidirectionally in either sense or antisense transcripts, which accumulate in RNA foci sequestering RNA-binding proteins (RBPs). (**C**) The HRE can be translated through a repeat-associated AUG-independent (RAN) translation mechanism producing toxic dipeptides (DPRs) generated from both the sense and antisense reading frames. Five DPRs have been described: glycine-alanine (GA), glycine-arginine (GR), proline-alanine (PA), proline-arginine (PR) and glycine-proline (GP).

**Figure 3 ijms-24-01130-f003:**
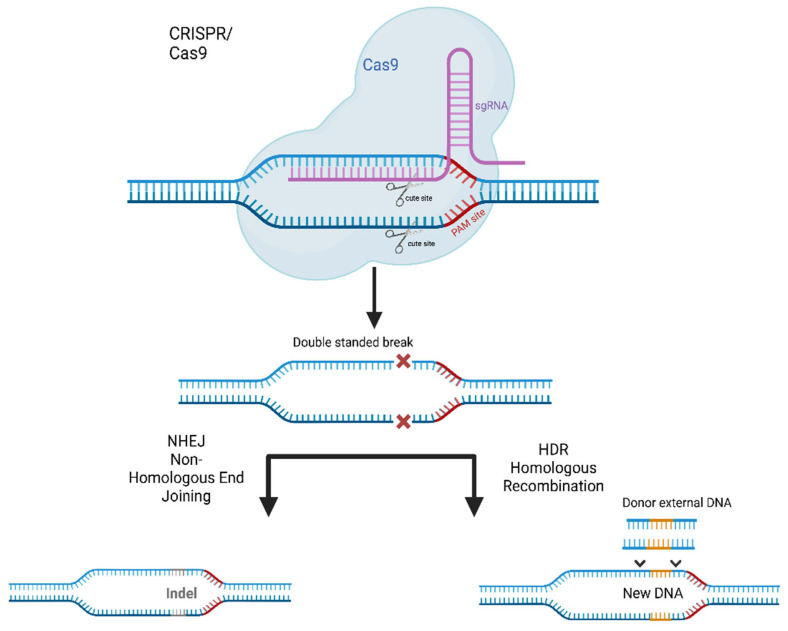
Techniques of CRISPR-Cas9 genome editing system.

**Table 1 ijms-24-01130-t001:** Comparison of approved drugs for SMA therapy.

Drug Name	Type	Mechanism of Action	Clinical Trials	Status
Nusinersen (Spinraza)	Antisense oligonucleotide	Binding SMN2 mRNA to modify splicing to increase SMN protein production	ENDEAR (Phase III) NURTURE (Phase II) CHERISH (Phase III) DEVOTE (Phase II/III) ONWARD (Phase III)	Approved by FDA in 2016
Onasemnogene abeparvovec (Zolgensma)	Gene replacement therapy	SMN1 gene delivery via adenovirus vector (AAV9)	START (Phase I) STR1VE (Phase III) NCT05089656 (Phase III)	Approved by FDA in 2019
Risdiplam (Evrysdi)	Small molecule	Binding directly to ESE2 of SMN2 transcript	FIREFISH (Phase II, III) SUNFISH (Phase II, III) JEWELFISH (Phase II) RAINBOWFISH (Phase II)	Approved by FDA in 2020

**Table 2 ijms-24-01130-t002:** Antisense oligonucleotides therapy Clinical Trials in Amyotrophic Lateral Sclerosis.

Gene Therapy	Agent	Phase	Citation	Date
SOD1		I	Miller et al. [104]	2013
	Tofersen (BIIB067)	I/II	Miller et al. [106]	2020
		III	NCT02623699	current
C9orf72	BIIB078	I	NCT03626012	2021
FUS	Jacifusen (ION363)	III	NCT04768972	current
ATXN2	BIIB100	I	NCT03945279	current

**Table 3 ijms-24-01130-t003:** Summary of research on gene targeting therapeutics mediated by viral vectors.

Gene Therapy	Model	Study	Major Findings	References
**AAV-RNAi**				
SOD1	SOD1^G93A^ mouse	Raoul et al., 2005	Delayed disease onset preserved motor neurons and axons, and improved motor function.	[96]
	SOD1^G93A^ mouse	Wang et al., 2008	It would improve the survival and function of motor neurons.	[89,130]
	SOD1^G93A^ mouse; SOD1^G37R^ mouse	Foust et al., 2013	Improved motor function, increased muscle mass and extended life expectancy in mice. Injections of neutral SOD1 mRNA reductions in the spinal cord of non-human primates.	[97]
	SOD1^G93A^ mouse	Borel et al., 2016	They obtained an extension of the median life expectancy of SOD1-G93A mice by 21% and preserved motor and respiratory function.	[100]
	SOD1^G93A^ mouse	Biferi et al., 2017	Increased survival in the tested mice and prevented the deterioration of neuromuscular function in them.	[101]
	Human SOD1-ALS phase I	Mueller et al., 2020	In 1 patient, clinical stabilization reduced the level of SOD1 in the spinal cord. Patient 2 without clinical benefits.	[134]
**Neuromuscular junction modulators**				
DOK7	SOD1^G93A^ mouse	Miyoshi et al., 2017	Improvement in motor activity and an extension of the life of the tested mice but no effect on motor neuron counts.	[135]
**Neurotrophic support**				
IGF	SOD1^G93A^ mouse	Lin et al., 2018	Significantly reduced the loss of the anterior lumbar spinal cord motor neurons and delayed muscle wasting in the tested mice. It delayed the onset of the disease and prolonged the lifespan of ALS mice.	[139]
**AAV-CRISPR**				
SOD1	SOD1^G93A^ mouse	Gaj et al., 2017	Reduced the mutant SOD1 protein > 2.5-fold in the thoracic and lumbar spine. In the final stage, the mice tested showed about 50% more motor neurons, and about 37% of the disease was delayed.	[148]
	SOD1^G93A^ mouse	Duan et al., 2020	Deletion of the SOD1 gene and the improvement in the life expectancy of the mice tested by 54.6%.	[149]
	Human SOD1^G93A^ transgenic mouse	Deng et al., 2021	No evidence of other diseases beyond the age of two in genome-edited mice.	[150]
C9orf72	Human FTD/ALS patient-derived iPSCs	Pribadi et al., 2016	Prevented RNA foci formation and promoter hypermethylation, two phenotypes of the C9orf72 mutation, but this did not significantly alter the expression of C9orf72 at the mRNA or protein level.	[151]
	Drosophila C9; human iPSC C9-ALS	Lopez-Gonzalez et al., 2019	Decreased activation of the apoptotic pathway and reduced nuclear foci.	[152]

## Data Availability

No new data were created or analyzed in this study. Data sharing is not applicable to this article.

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
