# Peer review of "Gene Therapy in ALS and SMA: Advances, Challenges and Perspectives"

_ijms, 2023, doi:10.3390/ijms24021130_

Round 1
Reviewer 1 Report (Previous Reviewer 2)
This paper is very special - molecular genetics of motor neuron diseases. The first part - on SMA, is well written, clearly understandable. But with those facts are readers well informed. The second part - on ALS - is difficult to read and to imagine various mechanisms of pathology, clinical trials, effects of experimental therapy. In SMA part: I think, more can be written about formation of spliceotome, silencing intron 7.
In ALS part: Only 5% patients begin with respiratory problems - reference needed.
5% patients with ALS live longer than 20 years - references needed.
Figure 4 - Some more detailed description will rational.
For genome editing - some scheme will be added
In desriptions of various clinical trials/ treatments some division of long collumnes and spacial setting will help the readers to concentrate on each separate problem.
Author Response
Response to Reviewer 1 Comments:
Dear Sir or Madam, thank you very much for the review our manuscript entitled: „ Gene therapy in ALS and SMA: advances, challenges and perspectives”
In response to your comment, we would like to thank you for appreciating our manuscript.
Comment 1
In SMA part: I think, more can be written about formation of spliceotome, silencing intron 7.
Revision and my comment
Added a paragraph in the SMA section on SMN2 splicing factors regulating exon 7 exclusion. (page 3, lines 116-130)
Comment 2
In ALS part: Only 5% patients begin with respiratory problems - reference needed.
Revision and my comment
We have added a reference to this sentence. We refer to the study by Niedermeyer et al. Respiratory Failure in Amyotrophic Lateral Sclerosis.
Comment 3
5% patients with ALS live longer than 20 years - references needed.
Revision and my comment
We have added a reference to this sentence. We refer to the review by Hulisz Amyotrophic Lateral Sclerosis: Disease State Overview
Comment 4
Figure 4 - Some more detailed description will rational.
Revision and my comment
We have extended the description to figure 4 with additional explanations about the pathomechanisms of C9orf72-associated ALS (page 13)
Comment 5
For genome editing - some scheme will be addedRevision and my comment
We created and added additional graphics on the techniques of CRISPR-Cas9 genome editing system (page 18).
Comment 6
In descriptions of various clinical trials/ treatments some division of long columns and special setting will help the readers to concentrate on each separate problem.
Revision and my comment
The results of clinical trials and studies on SMA and ALS therapies have been compiled and grouped into three new tables.
Table 1. Comparison of approved drugs for SMA therapy (pages 8,9).
Table 2. Antisense oligonucleotides therapy Clinical Trials in Amyotrophic Lateral Sclerosis (page 13).
Table 3. Summary of research on gene targeting therapeutics mediated by viral vectors (pages 17-18).
We do honestly hope that it will satisfy you and improve the quality of our work. Once again, we are very grateful for your review and remain open if you have any other remarks or suggestions that will make our work merit publication in „International Journal of Molecular Science”.

Reviewer 2 Report (New Reviewer)
In this manuscript the Authors reviewed early and more recent literature studies regarding gene therapy for the treatment of motor neuron diseases.
The issue addressed in this manuscript is of interest in the field of SMA and ALS.
However, from the reviewer’s point of view, major revisions, below reported, is needed to improve the article and make it acceptable for publication.
- The Authors should modify the Figure 1 enlarging the images. It is difficult to read.
- In the Paragraph 2.1 the Authors should update the reference list by adding the last papers on the issue focusing on the evidence after Nusinersen treatment in SMA patients (Bonanno et al., 2020, Biomedicines; Bonanno et al., 2022, Front Cell Neurosci).
- In the paragraph 3.1 the Authors should update and discuss the literature on the Trial of Antisense Oligonucleotide for SOD1 ALS (Miller et al., 2020and 2022; N Engl J Med.)
- The Authors should revise the Figure 3 and 4 because they did not show something new from the literature.
- The Authors should include a summary table of the different therapies described in the manuscript for SMA and ALS.
-In general the manuscript is difficult to read, the Authors should revise the work for a better and clear understanding. Specifically, the 3.1, 3.2, 3.4 paragraphs, the adeno-associated virus vectors paragraph.
Author Response
Response to Reviewer 2 Comments:
Dear Sir or Madam, thank you very much for the review our manuscript entitled: „ Gene therapy in ALS and SMA: advances, challenges and perspectives”
In response to your comment, we would like to thank you for appreciating our manuscript.
Comment 1
The Authors should modify the Figure 1 enlarging the images. It is difficult to read.
Revision and my comment
We have enlarged Figure 1 to make it easier to read
Comment 2
In the Paragraph 2.1 the Authors should update the reference list by adding the last papers on the issue focusing on the evidence after Nusinersen treatment in SMA patients (Bonanno et al., 2020, Biomedicines; Bonanno et al., 2022, Front Cell Neurosci).
Revision and my comment
Thank you for recommending the above references. Research by Bonnano et al. are described on page 5 lines 220-231.
Comment 3
In the paragraph 3.1 the Authors should update and discuss the literature on the Trial of Antisense Oligonucleotide for SOD1 ALS (Miller et al., 2020and 2022; N Engl J Med.)
Revision and my comment
Thank you for recommending the above references. Both studies are referenced in the section on antisense oligonucleotides (page 12, lines 522-533)
Comment 4
The Authors should revise the Figure 3 and 4 because they did not show something new from the literature.
Revision and my comment
We have removed Figure 3. We have also modified Figure 4, identifying a target for gene therapy in C9orf72-related fALS and extending the description with additional explanations on the pathomechanism of this disorder.
Comment 5
The Authors should include a summary table of the different therapies described in the manuscript for SMA and ALS.
Revision and my comment
The results of clinical trials and studies on SMA and ALS therapies have been compiled and grouped into three new tables.
Table 1. Comparison of approved drugs for SMA therapy (pages 8,9).
Table 2. Antisense oligonucleotides therapy Clinical Trials in Amyotrophic Lateral Sclerosis (page 13).
Table 3. Summary of research on gene targeting therapeutics mediated by viral vectors (pages 17-18).
Comment 6
In general, the manuscript is difficult to read, the Authors should revise the work for a better and clear understanding. Specifically, the 3.1, 3.2, 3.4 paragraphs, the adeno-associated virus vectors paragraph.
Revision and my comment
The indicated paragraphs have been reviewed and corrected. Two tables (Table 2 and Table 3) have been added, as well as Figure 5, to help the reader understand the studies described.
We do honestly hope that it will satisfy you and improve the quality of our work. Once again, we are very grateful for your review and remain open if you have any other remarks or suggestions that will make our work merit publication in „International Journal of Molecular Science”.

Reviewer 3 Report (New Reviewer)
Gene therapies hold great promise for neurological diseases, including spinal muscular atrophy (SMA) and amyotrophic lateral sclerosis (ALS). Covering elements of this topic, Lejman and colleagues have reviewed FDA-/EMA-approved and pre-clinical therapies for both SMA and ALS. The manuscript has potential, but in its current guise, it falls short of a publishable review due to a lack of coherency between title and content of the review, as well as many inaccurate wordings/definitions. I understand that a lot of work has gone into this article and it certainly has potential. To help improve the work, I have the following Major and Minor suggestions:
Major
· The title does not accurately reflect the content of the review. For example, the manuscript covers small molecules, which are not gene therapies. Furthermore, the review is restricted to SMA and ALS, and does not cover MNDs in general. I therefore suggest either a title change to focus on SMA and ALS therapeutics or a re-write of the abstract/content to match the title.
· The abstract needs to be drastically improved. For example, small molecules are not considered to be gene therapies; one must assume that SMA and ALS are classified as MNDs; “Examples can be found in the treatment of motor neuron diseases (MND)” is a poorly crafted sentence; SMA does not always result in premature death (Types 3 and 4); the sentence covering lines 20-22, does not make sense; it is unclear why Zolgensma is introduced with a separate identity of being a GRT.
· By far the weakest part of the manuscript is the Introduction. Errors are rife and the content does not reflect the focus of the article. For example, gene therapies do not necessarily “ensure a life-long therapeutic effect”; gene therapies are not always “based on repairing a defective gene” (e.g., neurotrophic factors); ASOs require “multiple doses” (e.g., Spinraza); “There are two classes of ASO” does not reflect Figure 1 and according to whom are there two classes?; how do ASOs “expand rapidly and widely”?; MNDs are poorly defined – what about peripheral neuropathies?; MNDs are described as arising due to “dysfunction of the upper motor neurons causing weakening of muscle function without pain symptoms or sensory disturbances” – upper motor neuron dysfunction is not a classic feature of SMA and sensory deficits are present in some forms of SMA and ALS; not all forms of SMA result in respiratory failure and death within 2-3 years. A complete re-write of the Introduction is required, with more careful descriptions/definitions.
Minor
· Lines 91-92: In its more severe forms, SMA is known to affect additional cell types and tissues (e.g., https://pubmed.ncbi.nlm.nih.gov/32644125/ and https://pubmed.ncbi.nlm.nih.gov/26506088/). This feature is relevant to treatment and should thus be introduced and discussed throughout the SMA section.
· As highlighted in the table, SMA types 3 and 4 does not affect life expectancy; hence the sentence on lines 93-95 is inaccurate.
· SMA severity is not solely “determined by the number of SMN2 copies”, as there are some patients with SMA type 2 that have the same number of SMN2 copies as patients with SMA type 3. On a population level – sure. Be more careful.
· “SMN2 is a gene identical to SMN1. The key difference is the C-T 104 substitution at position 6 in exon 7” – are they the same or different? Be more careful.
· The administration of Nusinersen does not just consist of four doses – please check the treatment regime.
· The ramifications of the following study should be addressed: https://www.nature.com/articles/s41593-021-00827-3
· “All patients survived.” More details, e.g., timescales, needed.
· There are now intrathecal trials with Zolgensma, please include.
· The importance of pre-symptomatic testing of therapies should be introduced. Moreover, please change “asymptomatic” for “pre-symptomatic” – the former is not necessarily correct.
· The section entitled, “Other Clinical Trials” for SMA includes details of experiments not related to clinical trials and should be amended.
· Targeting of SMN-AS1 could be introduced: https://pubmed.ncbi.nlm.nih.gov/28017471/
· The sentence stating that ALS “affects about 16,000 people” needs to be clarified and referenced.
· Lines 308-309: the sentence structure suggests that the frequency of both sALS and fALS ranges fro 5-10%, which does not make sense.
· Riluzole and edaravone do not necessarily “improve the quality of life of patients.” It would be more accurate to say that they marginally increase survival time.
· The meaning of “12% of family patients” is unclear.
· The following important ALS clinical trial paper is missing: https://pubmed.ncbi.nlm.nih.gov/36129998/
· The ability of viral vectors to cross the blood-brain barrier is mentioned. Please provie references and be more cautious in this discussion – is the crossing at particular ages, or is it know that this occurs also in adults?
· “…the AAV9 serotype, which can be highly express in neurons…” does not make sense.
· Please provide more informative figure and table legends.
· The table is not really needed, and Figures 2 and 3 are not informative to the SMA or ALS fields. Figures more relevant to the topic of the review would be more appealing. For example, images depicting the different types of therapeutic of SMA and then ALS.
· The reference list needs to be formalised; there is no consistency and several references include only hyperlinks.
Author Response
Response to Reviewer 3 Comments:
Dear Sir or Madam, thank you very much for the review our manuscript entitled: „ Gene therapy in ALS and SMA: advances, challenges and perspectives”
In response to your comment, we would like to thank you for appreciating our manuscript.
Comment 1
The title does not accurately reflect the content of the review. For example, the manuscript covers small molecules, which are not gene therapies. Furthermore, the review is restricted to SMA and ALS, and does not cover MNDs in general. I therefore suggest either a title change to focus on SMA and ALS therapeutics or a re-write of the abstract/content to match the title.
Revision and my comment
We changed the title to reflect the content of our literature review.
Comment 2
The abstract needs to be drastically improved. For example, small molecules are not considered to be gene therapies; one must assume that SMA and ALS are classified as MNDs; “Examples can be found in the treatment of motor neuron diseases (MND)” is a poorly crafted sentence; SMA does not always result in premature death (Types 3 and 4); the sentence covering lines 20-22, does not make sense; it is unclear why Zolgensma is introduced with a separate identity of being a GRT.
Revision and my comment
The abstract has been rewritten considering the above-mentioned remarks. The sentences quoted above have been removed or corrected.
Comment 3
By far the weakest part of the manuscript is the Introduction. Errors are rife and the content does not reflect the focus of the article. For example, gene therapies do not necessarily “ensure a life-long therapeutic effect”; gene therapies are not always “based on repairing a defective gene” (e.g., neurotrophic factors); ASOs require “multiple doses” (e.g., Spinraza); “There are two classes of ASO” does not reflect Figure 1 and according to whom are there two classes?; how do ASOs “expand rapidly and widely”?; MNDs are poorly defined – what about peripheral neuropathies?; MNDs are described as arising due to “dysfunction of the upper motor neurons causing weakening of muscle function without pain symptoms or sensory disturbances” – upper motor neuron dysfunction is not a classic feature of SMA and sensory deficits are present in some forms of SMA and ALS; not all forms of SMA result in respiratory failure and death within 2-3 years. A complete re-write of the Introduction is required, with more careful descriptions/definitions.
Revision and my comment
The introduction has been revised and thoroughly improved. We checked the references again, and the information used was described with attention to details and factual correctness.
Comment 4
Lines 91-92: In its more severe forms, SMA is known to affect additional cell types and tissues (e.g., https://pubmed.ncbi.nlm.nih.gov/32644125/ and https://pubmed.ncbi.nlm.nih.gov/26506088/). This feature is relevant to treatment and should thus be introduced and discussed throughout the SMA section.
Revision and my comment
Thank you for recommending the above-mentioned research. Their descriptions have been added on page 3 in lines 96-101
Comment 5
As highlighted in the table, SMA types 3 and 4 does not affect life expectancy; hence the sentence on lines 93-95 is inaccurate.
Revision and my comment
The description of the clinical picture of SMA has been clarified. It has been emphasized that death affects only the most severe cases (page 3 lines 101-106).
Comment 6
SMA severity is not solely “determined by the number of SMN2 copies”, as there are some patients with SMA type 2 that have the same number of SMN2 copies as patients with SMA type 3. On a population level – sure. Be more careful.
Revision and my comment
This misleading sentence has been removed.
Comment 7
“SMN2 is a gene identical to SMN1. The key difference is the C-T 104 substitution at position 6 in exon 7” – are they the same or different? Be more careful.
Revision and my comment
We have corrected this sentence by noting that SMN1 and SMN2 are nearly identical (page 3 lines 101-102).
Comment 8
The administration of Nusinersen does not just consist of four doses – please check the treatment regime.
Revision and my comment
The nusinersen dosing description has been changed to reflect the treatment regime (page 4, lines 169-172).
Comment 9
The ramifications of the following study should be addressed: https://www.nature.com/articles/s41593-021-00827-3
Revision and my comment
Thank you for recommending the above-mentioned research. Their descriptions have been added on page 7 in lines 97-114
Comment 10
“All patients survived.” More details, e.g., timescales, needed.
Revision and my comment
We have corrected this sentence (page 5 lines 189-192)
Comment 11
There are now intrathecal trials with Zolgensma, please include.
Revision and my comment
We have included attempts at intrathecal administration of Zolgensma in the paragraph on page 7 lines 293-297
Comment 12
The importance of pre-symptomatic testing of therapies should be introduced. Moreover, please change “asymptomatic” for “pre-symptomatic” – the former is not necessarily correct.
Revision and my comment
Correction applied page 8 lines 368-371.
Comment 13
The section entitled, “Other Clinical Trials” for SMA includes details of experiments not related to clinical trials and should be amended.
Revision and my comment
We changed the title of the section to “Challenges and future directions” page 9 line 377
Comment 14
Targeting of SMN-AS1 could be introduced: https://pubmed.ncbi.nlm.nih.gov/28017471/
Revision and my comment
Thank you for recommending the above-mentioned research. Their descriptions have been added on page 9 lines 384-391
Comment 15
The sentence stating that ALS “affects about 16,000 people” needs to be clarified and referenced.
Revision and my comment
We changed the data on the occurrence of ALS on the basis of research by Mehta et al. and Marin et al. page 9 lines 408-412
Comment 16
Lines 308-309: the sentence structure suggests that the frequency of both sALS and fALS ranges fro 5-10%, which does not make sense.
Revision and my comment
Correction applied page 9 lines 415-417.
Comment 17
Riluzole and edaravone do not necessarily “improve the quality of life of patients.” It would be more accurate to say that they marginally increase survival time.
Revision and my comment
Correction applied page 10 lines 429-430
Comment 18
The meaning of “12% of family patients” is unclear.
Revision and my comment
Correction applied page 10 lines 461-463
Comment 19
The following important ALS clinical trial paper is missing: https://pubmed.ncbi.nlm.nih.gov/36129998/
Revision and my comment
Thank you for recommending the above-mentioned research. Their descriptions have been added on page 12, lines 527-533
Comment 20
The ability of viral vectors to cross the blood-brain barrier is mentioned. Please provie references and be more cautious in this discussion – is the crossing at particular ages, or is it know that this occurs also in adults?
Revision and my comment
The study by Duque et al. has been added to the references (page 13, lines 568-571).
Comment 21
“…the AAV9 serotype, which can be highly express in neurons…” does not make sense.
Revision and my comment
Correction applied page 6 lines 255-258
Comment 22
Please provide more informative figure and table legends.
Revision and my comment
Three tables have been added, as well as Figure 5, to help the reader understand the studies described. We have also modified Figure 4, identifying a target for gene therapy in C9orf72-related fALS and extending the description with additional explanations on the pathomechanism of this disorder.
Comment 23
The table is not really needed, and Figures 2 and 3 are not informative to the SMA or ALS fields. Figures more relevant to the topic of the review would be more appealing. For example, images depicting the different types of therapeutic of SMA and then ALS.
Revision and my comment
Figures 2 and 3 as well as Table 1 have been removed.
Comment 24
The reference list needs to be formalised; there is no consistency and several references include only hyperlinks.
Revision and my comment
Care was taken to better organize, formalize and standardize the list of references.
We do honestly hope that it will satisfy you and improve the quality of our work. Once again, we are very grateful for your review and remain open if you have any other remarks or suggestions that will make our work merit publication in „International Journal of Molecular Science”.

Reviewer 4 Report (New Reviewer)
In this review, the authors describe the use of various oligonucleotides, gene therapy agents, and genome editing in the treatment of SMA and familial ALS. These topics have attracted enormous attention from both clinicians and scientists in recent years. I should first note that dozens of reviews on SMA therapeutics have been published in the last few years, several of which are excellent. The main limitation of the present work is that it does not bring any new information or insight in this regard. In addition, the manuscript also contains a significant number of scientific inaccuracies, and it is lacking in both organization and coverage of recent developments in the field:
* Scientific content and accuracy: The manuscript contains several scientific inaccuracies. These appear small and correctable when taken individually, but their large number is very problematic. I am presenting here a few examples, taken primarily from the first pages:
- "Gene therapy [...] ensures a life-long therapeutic effect": this is at best a hope, and not yet proven. Loss of viral genomes, silencing of promoters, and other factors can all affect long-term gene expression.
- "repairing a defective gene": This implies a form of gene editing, which is not what the authors are describing.
- "There are two chemical classes of ASOs": This is incorrect because (a) there are more than two ASO chemistries, (b) the following sentences discuss mechanisms of ASOs, not chemical classes, and (c) there are more than two mechanisms of action of ASOs – indeed nusinersen itself is an example of an ASO that acts through through neither RNA interference, nor RNaseH.
- "Dysfunction of the upper motor neurons": SMA is an exclusively lower motor neuron disorder, while ALS involves both upper and lower motor neurons.
- The similarities between SMA and ALS are over-stated, given that the two disorders have a very different underlying pathogenesis. The blanket statement that both SMA and ALS lead to "death within 2-3 years" is incorrect.
- "[Nusinersen] is administered intrathecally in 4 doses": There are 4 loading doses, followed by regular maintenance infusions.
- "It uses AAV9 adenoviruses to deliver recombinant DNA complementary to SMN1": (a) adeno-associated viruses are not adenoviruses (b) the SMN transgene delivered by onasemnogene abeparvovec is not "complementary to SMN1".
- "Low dose prednisone": The prednisone dose used for gene replacement is generally 1mg/kg, which is not considered low-dose.
- There are no longer any age limitations on the FDA approval of risdiplam.
- Serious drug-related AEs with risdiplam: The SAEs listed on lines 276-277 have indeed been reported in the risdiplam clinical trials, but I am not aware of them having been adjudicated as drug-related. In fact, Mercuri et al [PMID 35837793] explicitly says that no SAEs were judged to be drug-related.
- "Research trials of PMO in the treatment of SMA are ongoing": I understand this to mean that there are ongoing clinical trials of PMOs in SMA, but I am unaware of any such trials.
- "Only 2 drugs are used to improve the quality of life": The authors fail to mention dextromethorphan HBr and quinidine sulfate (Nudexta), as well as the many other drugs used for symptom management in ALS.
- "This specificity of AAV delivery, however, can be achieved by using a specific promoter": The authors probably mean that the specificity of gene expression can be enhanced by use of an organ-specific promoter.
- "Although vectors are designed in a way that should make replication difficult and avoid virulence, this is not always completely successful": I am uncertain what studies the authors may be referring to. Nearly all recent human gene therapy clinical trials have used AAV vectors. The genomes of these engineered vectors do not code for any AAV proteins and are thus incapable of replication.
* Overall content and organisation:
- The introduction jumps back and forth between ASOs and gene therapy, making it difficult to follow. The first paragraphs are a general description of gene therapy and RNA therapeutics, without making a clear connection to motor neuron disorders.
- The meaning of the term "gene therapy" shifts through the manuscript, sometimes including RNA-directed therapies and sometimes only gene replacement.
- A full half-page is dedicated to discussing the RULM and 6MWT, but these are essentially never mentioned again. By contrast, no information is provided on the scales that actually formed the primary endoints in the SMA and ALS trials. The discussion of the 6MWT makes no mention of the significant flaws that have led it to be de-emphasised as a primary endpoint in neuromuscular disease clinical trials.
- Background information on AAV and other viral vectors is presented in several places in the manuscript and the main paragraph dealing with this is located in the section on ALS, i.e. after the discussion of onasemnogene abeparvovec.
- The discussion on the pros and cons of lentiviral and AAV vectors lacks key elements of information, including their packaging capacities and the oncogenicity of lentiviral vectors.
- In the section on ALS, there are several jumps between different forms of fALS and it is sometimes unclear what specific gene a given sentence may refer to.
* Several important recent and ongoing developments in MND therapeutics are not discussed in the manuscript:
- Recent and ongoing clinical trials in SMA: There is no mention of SRK-015, reldesemtiv, intrathecal onasemnogene abeparvovec, high-dose nusinersen, etc.
- Other contemporary clinical issues in SMA care: The manuscript likewise makes no mention of issues such as the relative efficacy of the 3 FDA-approved agents, the role of combination therapy, the reasons why some patients may respond better than others, etc.
- Post-marketing experience with SMA drugs: This is not discussed in any significant way, despite the fact that published post-marketing data in SMA contains important lessons about safety, timing of therapy, and several of the points raised above.
- Oral edaravone: data supporting its efficacy has in fact been presented, and the drug is now FDA-approved
- Tofersen: The recent clinical trial data published by Miler et al. in NEJM is not discussed.
* English language: The manuscript makes use in several places of imprecise terminology that gives it an un-scientific tone. The following are a few examples of this, but not an exhaustive list: "conformities", "sitting disability", "the SMN1 gene is damaged", "exceed the blood-brain barrier", "the Zolgensma activity", "convulsions", "the peak of viral vectors' potential", "as it turns out", "cracks" in DNA, etc. There are also a handful of grammar mistakes.
Author Response
Response to Reviewer 4 Comments:
Dear Sir or Madam, thank you very much for the review our manuscript entitled: „ Gene therapy in ALS and SMA: advances, challenges and perspectives”
In response to your comment, we would like to thank you for appreciating our manuscript.
Comment 1
"Gene therapy [...] ensures a life-long therapeutic effect": this is at best a hope, and not yet proven. Loss of viral genomes, silencing of promoters, and other factors can all affect long-term gene expression.
Revision and my comment
Correction applied (page 1 line 32)
Comment 2
"repairing a defective gene": This implies a form of gene editing, which is not what the authors are describing.
Revision and my comment
Correction applied (page 1 lines 32-34)
Comment 3
- "There are two chemical classes of ASOs": This is incorrect because (a) there are more than two ASO chemistries, (b) the following sentences discuss mechanisms of ASOs, not chemical classes, and (c) there are more than two mechanisms of action of ASOs – indeed nusinersen itself is an example of an ASO that acts through through neither RNA interference, nor RNaseH.
Revision and my comment
Correction applied (page 1 line 41)
Comment 4
"Dysfunction of the upper motor neurons": SMA is an exclusively lower motor neuron disorder, while ALS involves both upper and lower motor neurons
Revision and my comment
This misleading sentence has been removed.
Comment 5
The similarities between SMA and ALS are over-stated, given that the two disorders have a very different underlying pathogenesis. The blanket statement that both SMA and ALS lead to "death within 2-3 years" is incorrect.
Revision and my comment
Correction applied (pages 1-2 lines 76-78)
Comment 6
"[Nusinersen] is administered intrathecally in 4 doses": There are 4 loading doses, followed by regular maintenance infusions.
Revision and my comment
The nusinersen dosing description has been changed to reflect the treatment regime (page 4, lines 169-172).
Comment 7
"It uses AAV9 adenoviruses to deliver recombinant DNA complementary to SMN1": (a) adeno-associated viruses are not adenoviruses (b) the SMN transgene delivered by onasemnogene abeparvovec is not "complementary to SMN1".
Revision and my comment
Correction applied (pages 5-6 lines 236-237)
Comment 8
"Low dose prednisone": The prednisone dose used for gene replacement is generally 1mg/kg, which is not considered low-dose.
Revision and my comment
Correction applied (page 7 lines 301-302)
Comment 9
There are no longer any age limitations on the FDA approval of risdiplam
Revision and my comment
We have added information about this on page 7 lines 318-319
Comment 10
Serious drug-related AEs with risdiplam: The SAEs listed on lines 276-277 have indeed been reported in the risdiplam clinical trials, but I am not aware of them having been adjudicated as drug-related. In fact, Mercuri et al [PMID 35837793] explicitly says that no SAEs were judged to be drug-related.
Revision and my comment
Correction applied (page 8 lines 373-375)
Comment 11
"Research trials of PMO in the treatment of SMA are ongoing": I understand this to mean that there are ongoing clinical trials of PMOs in SMA, but I am unaware of any such trials.
Revision and my comment
This misleading sentence has been removed.
Comment 12
"Only 2 drugs are used to improve the quality of life": The authors fail to mention dextromethorphan HBr and quinidine sulfate (Nudexta), as well as the many other drugs used for symptom management in ALS.
Revision and my comment
Correction applied (page 10 lines 430-431)
Comment 13
"This specificity of AAV delivery, however, can be achieved by using a specific promoter": The authors probably mean that the specificity of gene expression can be enhanced by use of an organ-specific promoter.
Revision and my comment
Correction applied (page 6 lines 262-263)
Comment 14
"Although vectors are designed in a way that should make replication difficult and avoid virulence, this is not always completely successful": I am uncertain what studies the authors may be referring to. Nearly all recent human gene therapy clinical trials have used AAV vectors. The genomes of these engineered vectors do not code for any AAV proteins and are thus incapable of replication.
Revision and my comment
Correction applied (page 17 lines 746-749)
Comment 15
The introduction jumps back and forth between ASOs and gene therapy, making it difficult to follow. The first paragraphs are a general description of gene therapy and RNA therapeutics, without making a clear connection to motor neuron disorders.
Revision and my comment
The introduction has been revised and thoroughly improved. We checked the references again, and the information used was described with attention to details and factual correctness.
Comment 16
The meaning of the term "gene therapy" shifts through the manuscript, sometimes including RNA-directed therapies and sometimes only gene replacement.
Revision and my comment
With this remark in mind, the entire manuscript has been revised and inaccurate terms have been corrected.
Comment 17
A full half-page is dedicated to discussing the RULM and 6MWT, but these are essentially never mentioned again. By contrast, no information is provided on the scales that actually formed the primary endoints in the SMA and ALS trials. The discussion of the 6MWT makes no mention of the significant flaws that have led it to be de-emphasised as a primary endpoint in neuromuscular disease clinical trials.
Revision and my comment
The paragraph on motor skills assessment scales has been limited only to those mentioned in the studies we describe and moved to the end of the chapter on SMA page 4 lines 131-158.
Comment 18
Background information on AAV and other viral vectors is presented in several places in the manuscript and the main paragraph dealing with this is located in the section on ALS, i.e. after the discussion of onasemnogene abeparvovec.
Revision and my comment
General information on AAV has been moved to the ononasemnogene abeparvovec section (page 6 lines 241-267).
Comment 19
The discussion on the pros and cons of lentiviral and AAV vectors lacks key elements of information, including their packaging capacities and the oncogenicity of lentiviral vectors.
Revision and my comment
The section on the use of viral vectors in ALS has undergone significant revisions. New references and missing information have been added.
Comment 20
In the section on ALS, there are several jumps between different forms of fALS and it is sometimes unclear what specific gene a given sentence may refer to.
Revision and my comment
We have added Table 2 and Table 3 to clarify which gene is being targeted in a given study or clinical trial.
Comment 21
Recent and ongoing clinical trials in SMA: There is no mention of SRK-015, reldesemtiv, intrathecal onasemnogene abeparvovec, high-dose nusinersen, etc.
Revision and my comment
We have included clinical trials of high-dose nusinersen page lines 206-219 and attempts at intrathecal administration of Zolgensma in the paragraph on page 7 lines 293-297.
Comment 22
Other contemporary clinical issues in SMA care: The manuscript likewise makes no mention of issues such as the relative efficacy of the 3 FDA-approved agents, the role of combination therapy, the reasons why some patients may respond better than others, etc. Post-marketing experience with SMA drugs: This is not discussed in any significant way, despite the fact that published post-marketing data in SMA contains important lessons about safety, timing of therapy, and several of the points raised above.
Revision and my comment
We added studies on the use of miRNAs and cytokines of the Th1/Th17 pathway as potential biomarkers of response to nusinersen treatment page 5 lines 220-230.
Comment 23
Oral edaravone: data supporting its efficacy has in fact been presented, and the drug is now FDA-approved.
Revision and my comment
We have developed information about endarovone page 10 lines 440-448
Comment 25
Tofersen: The recent clinical trial data published by Miler et al. in NEJM is not discussed.
Revision and my comment
Thank you for recommending the above-mentioned research. Their descriptions have been added on page 12, lines 527-533
Comment 26
The manuscript makes use in several places of imprecise terminology that gives it an un-scientific tone. The following are a few examples of this, but not an exhaustive list: "conformities", "sitting disability", "the SMN1 gene is damaged", "exceed the blood-brain barrier", "the Zolgensma activity", "convulsions", "the peak of viral vectors' potential", "as it turns out", "cracks" in DNA, etc. There are also a handful of grammar mistakes.
Revision and my comment
The manuscript has been checked and corrected by a native speaker
We do honestly hope that it will satisfy you and improve the quality of our work. Once again, we are very grateful for your review and remain open if you have any other remarks or suggestions that will make our work merit publication in „International Journal of Molecular Science”.

Round 2
Reviewer 2 Report (New Reviewer)
The authors have satisfactorily answered to all comments
Author Response
Thank you very much for your comments.
Best regards
Monika Lejman
Reviewer 3 Report (New Reviewer)
The authors have done a good job at responding to my comments and the manuscript is now much better.
Author Response
Thank you very much for your comments
Best regards
Monika Lejman
Reviewer 4 Report (New Reviewer)
In this revised submission, the authors have addressed some of my previous concerns. Nonetheless, the manuscript continues to suffer from a number of factual inaccuracies and English mistakes that need to be corrected.
- Lines 45-46: I do not understand the meaning of this sentence.
- Line 51: It is misleading to describe only two "main" classes of ASOs in the introduction when the ASO most extensively discussed in the manuscript (nusinersen) does not fall in either of these classes.
- Lines 164-191: This paragraph contains a number of English language mistakes, including "while the second examines by performing 8 positions by the patient", "the scale allows you to observe", "turning over from toe to stomach", etc.
- Line 171+: The description of the HINE scale is incorrect; each item is scored between 0-3. The neurological examination consists of 5 (not 2) domains and the infant does not need to "perform" any positions. Although milestones are recorded, they are not scored and it is incorrect to say that "The scale allows you to observe the child's development based on the achievement of individual milestones."
- Lines 185-191: The authors describe how children with SMA can have a wide range of scores on the HFMSE. They seem to present this as a limitation, while the ability of the scale to capture such a wide range of outcomes is in fact a desirable feature.
- Line 243: "he examines" should be corrected
- Lines 255-265: Some introductory statement should be provided to explain why these studies are being described here.
- Line 299: Onasemnogene abeparvovec does not deliver wild-type SMN1 gene, since the gene has a synthetic promoter, does not include introns, and may also have other sequence changes (e.g. codon optimisation)
- Line 305: The authors discuss the rep and cap genes, but fail to mention that these are not included in any AAV-based therapeutics.
- Line 311: The phrase "the ability of the immune response" should be revised
- Line 318: This should say "leads to high expression", not "contributes"
- Line 377: The phrase "turned out to be" should be revised. Other variants of this phrase in the manuscript (e.g. it turns out) should also be eliminated.
- Lines 500-501: The phrasing here is quite strange. The authors should more plainly state that riluzole and edaravone are the only two agents shown to slow down disease progression in ALS.
- Line 587: It is unclear why the word "administered" was removed.
In addition to the above English language mistakes, I would also note that the manuscript frequently makes use of phrases, words, and sentence structures that are "awkward" in English or that are not used in scientific writing. While it is generally possible to understand the intended meaning, these give the manuscript an overall colloquial tone, rather than a precise, scientific and authoritative tone. I am providing here a few examples, but there are multiple instances of such phrases on nearly every page:
- For nusinersen to be approved, it had to show effectiveness in several studies
- Below we present the promising results of several clinical trials that have approved the use of nusinersen.
- Apart from non-vector gene therapy, viral carriers have also found their place in clinical trials of ALS. Moreover, it turns out that they may represent a promising future.
- In the context of ALS gene therapy, the AAV9 serotype appears to be promising.
- If their results prove to be positive, it could have a significant impact on the further development of gene therapy in ALS.
- It is worth mentioning that modulation of the SOD1 gene activity is not the end of the potential of viral vectors in ALS
Author Response
In this revised submission, the authors have addressed some of my previous concerns. Nonetheless, the manuscript continues to suffer from a number of factual inaccuracies and English mistakes that need to be corrected.
- Lines 45-46: I do not understand the meaning of this sentence. - It has been corrected.
“ASO-based therapies are currently used to treat a variety of conditions including: mipomersen for homozygous familial hypercholesterolemia, fomivirsen for cytomegalovirus retinitis, miravirsen for hepatitis C virus (HCV), eteplirsen for Duchenne muscular dystrophy and Spinraza for spinal atrophy muscles [2, 4, 5].They are used as a promising treatment for diseases in various fields - mipomersen for homozygous familial hypercholesterolemia, fomivirsen for cytomegalovirus retinitis, miravirsen for the hepatitis C virus (HCV), eteplirsen for Duchenne muscular dystrophy, and Spinraza for spinal muscular atrophy ​[2,4,5]​.”
- Line 51: It is misleading to describe only two "main" classes of ASOs in the introduction when the ASO most extensively discussed in the manuscript (nusinersen) does not fall in either of these classes. - It has been corrected.
“There are different classes of ASOs, some of which are: first-class single-stranded ASOs degrade specific RNA or modulate its metabolism by the enzyme RNaseH, and second-class double-stranded synthetic oligonucleotides degrade RNA via an RNA-induced silencing complex (RISC) [3]​.
There are two main classes of ASO. The first class- single-stranded ASOs degrade specific RNA or modulate its metabolism by the RNaseH enzyme. Second class- double-stranded synthetic oligonucleotides degrade RNA via the RNA-induced silencing complex (RISC) ​[3]​.”
- Lines 164-191: This paragraph contains a number of English language mistakes, including "while the second examines by performing 8 positions by the patient", "the scale allows you to observe", "turning over from toe to stomach", etc. - It has been corrected.
“The HINE is a clinical neurological assessment test for infants 2 to 24 months of age whose cutoff scores for predicting walking or sitting ability can provide important prognostic information for future motor development. The test consists of 3 parts: (1) neurological examination (26 items, scored on a scale from 0 to 3; total points - 78) assessing cranial nerve function, posture, movements, tension, reflexes and reactions, (2) motor milestones (8 items unrated) and (3) behavior (3 items unrated). The total score can be classified as optimal (>73) or suboptimal. It also has modifications to assess the motor function of people with milder types of SMA [34]. HINE scale consists of two five parts. The first consists in observing changes in the neurological examination, while the second examines by performing 8 positions by the patient. Points range from 0 to 26 with subpoints ranging from 0-4. The scale allows you to observe the child's development based on the achievement of individual milestones. It is quite often used in clinical practice and easy to implement. The Motor Function Measure (MFM) is another validated tool for the numerical measurement of motor skills of patients with SMA type 2 and 3. The MFM tests both fine and gross motor skills, i.e., lifting the head, changing the position from lying to sitting, turning over from toe to stomach and holding coins, tearing a piece of paper or drawing loops. Each task is scored from 0 to 3 points ​[25]​. “
- Line 171+: The description of the HINE scale is incorrect; each item is scored between 0-3. The neurological examination consists of 5 (not 2) domains and the infant does not need to "perform" any positions. Although milestones are recorded, they are not scored and it is incorrect to say that "The scale allows you to observe the child's development based on the achievement of individual milestones."- It has been corrected.
“The HINE is a clinical neurological assessment test for infants 2 to 24 months of age whose cutoff scores for predicting walking or sitting ability can provide important prognostic information for future motor development. The test consists of 3 parts: (1) neurological examination (26 items, scored on a scale from 0 to 3; total points - 78) assessing cranial nerve function, posture, movements, tension, reflexes and reactions, (2) motor milestones (8 items unrated) and (3) behavior (3 items unrated). The total score can be classified as optimal (>73) or suboptimal. It also has modifications to assess the motor function of people with milder types of SMA [34]. HINE scale consists of two five parts. The first consists in observing changes in the neurological examination, while the second examines by performing 8 positions by the patient. Points range from 0 to 26 with subpoints ranging from 0-4. The scale allows you to observe the child's development based on the achievement of individual milestones. It is quite often used in clinical practice and easy to implement. The Motor Function Measure (MFM) is another validated tool for the numerical measurement of motor skills of patients with SMA type 2 and 3. The MFM tests both fine and gross motor skills, i.e., lifting the head, changing the position from lying to sitting, turning over from toe to stomach and holding coins, tearing a piece of paper or drawing loops. Each task is scored from 0 to 3 points ​[25]​. “
- Lines 185-191: The authors describe how children with SMA can have a wide range of scores on the HFMSE. They seem to present this as a limitation, while the ability of the scale to capture such a wide range of outcomes is in fact a desirable feature. - It has been removed.
“Another tool used to assess patients with SMA types 2 and 3 is the Hammersmith Extended Functional Scale (HFMSE). The assessment elements included in the HFMSE have proven to be extremely useful in clinical practice as a rehabilitation assessment tool, as well as in clinical trials to determine disease progression. One of the challenges of using this scale is that patients with spinal muscular atrophy are a clinically very heterogeneous group; and even if we restrict ourselves to the type 2 and type 3 phenotypes whose functional domains are covered by the HFMSE, the clinical picture still ranges from sedentary patients with only a few points on the scale to patients who can perform almost all of the 33 items of the scale. The maximum number of points to be obtained is 66 ​[26]​. “
- Line 243: "he examines" should be corrected. - It has been corrected.
“In part A, he examines patients with late onset of SMA symptoms (> 6 months). Part A includes patients with late-onset SMA symptoms (>6 months).”
- Lines 255-265: Some introductory statement should be provided to explain why these studies are being described here. - It has been added.
“In addition, there is an emphasis in more recent literature to discover new biological markers that can be used to assess patient progress, and increasingly, research suggests that miRNAs may act as primary modulators of SMN-mediated molecular pathways. Moreover, inflammatory molecules may represent new potential therapeutic targets as well as reliable biomarkers useful for patient stratification, predicting disease progression and monitoring response to therapies, and consequently better treatment of SMA patients. At this point, it is also worth mentioning the study by Bonanno et al. ​[34]​, which showed that nusinersen also decreased skeletal muscle-specific miRNA levels in 21 patients with SMA types II and III, associated with the pathogenic process in neuromuscular diseases. Downregulation of these miRNAs correlated with improved motor function of patients evaluated by HFMSE ​[34]​. In another study of 21 pediatric patients with SMA types 1, 2, and 3; and 12 adults with SMA types 2 and 3 Bonanno et al. ​[35]​ showed that nusinersen therapy may have a beneficial effect on the peripheral immune system. After 6 months of treatment, a decrease in the level of pro-inflammatory cytokines of the Th1/Th17 pathway was noted in the serum of patients. Interestingly, the above studies identified miR-133a and IL-23 molecules as potential predictive biomarkers of nusinersen therapy, and IL-10 as a potential biomarker for on-treatment monitoring ​[34,35]​. “
- Line 299: Onasemnogene abeparvovec does not deliver wild-type SMN1 gene, since the gene has a synthetic promoter, does not include introns, and may also have other sequence changes (e.g. codon optimisation). - It has been corrected.
“It uses a non-replicating adeno-associated virus capsid (scAAV9) to deliver transgene under a ubiquitous promoter wild-type SMN1 gene SMN to motor neuron cells. It is worth noting that, in contrast to nusinersen, Zolgensma crosses the blood-brain barrier and one administration per 1-hour intravenous infusion is sufficient for the systemic expression of the SMN protein ​[28]​.“
- Line 305: The authors discuss the rep and cap genes, but fail to mention that these are not included in any AAV-based therapeutics. - The sentence has been added:
“In contrast, the rep gene encodes proteins involved in AAV replication, transcription, integration and encapsidation ​[37]​. However, rep and cap genes are not included in any AAV-based therapeutics yet.”
- Line 311: The phrase "the ability of the immune response" should be revised. - It has been corrected.
“Among themselves, they show differences in cell tropism (depending on the type of capsid surface proteins), differences in transduction efficiency and differences in the ability of the immune response in immune response capacity.”
- Line 318: This should say "leads to high expression", not "contributes". - It has been corrected.
“In this context, we can distinguish the AAV2 serotype, which is specific for cerebral endothelial cells, or the AAV9 serotype, which after systemic administration contributes leads to high expression in neurons of the motor cortex, cerebellum, substantia nigra and cervical spinal cord ​[7,8,40]​.“
- Line 377: The phrase "turned out to be" should be revised. Other variants of this phrase in the manuscript (e.g. it turns out) should also be eliminated. - The phrase "turned out to be" has been removed. Similarly, the phrase "turns out” has also been removed.
“Good results in patients after administration of the currently approved nusinersen, Zolgensma or risdiplam give an optimistic perspective for further studies. It was found to be of particular importance turned out to be particularly significant in the pathology of the same group - ALS, in which several drugs are currently undergoing clinical.”
“This turned out proved to be a significant difference compared to the historical cohort in which only 8% of patients survived the threshold of 20 months of life without constant ventilation.”
“Apart from non-vector gene therapy, viral carriers have also found their place in clinical trials of ALS. Moreover, it turns out that they may represent a promising future. So far, gene delivery to the central nervous system has been limited for most therapeutic molecules due to the existence of the blood-brain barrier.”
- Lines 500-501: The phrasing here is quite strange. The authors should more plainly state that riluzole and edaravone are the only two agents shown to slow down disease progression in ALS. - The sentence has been added and the whole part of the text has been changed.
“Gene therapy for two forms of the disease - one caused by mutations in the C9orf72 gene and the form related to superoxide dismutase 1 (SOD1) Cu / Zn - may be a promising treatment therapy in the case of failure of pharmacological approaches.
Gene therapy with riluzole and edaravone - the only two known agents shown to slow down disease progression in ALS - for two forms of the disease - one caused by mutations in the C9orf72 gene and the form related to superoxide dismutase 1 (SOD1) Cu / Zn - may be a promising treatment therapy in the case of failure of pharmacological approaches.”
- Line 587: It is unclear why the word "administered" was removed. - The word “administered” has been added.
“In another attempt, Rizvanov et al. ​[99]​ administered siRNA locally to the sciatic nerve of the mouse model.”
In addition to the above English language mistakes, I would also note that the manuscript frequently makes use of phrases, words, and sentence structures that are "awkward" in English or that are not used in scientific writing. While it is generally possible to understand the intended meaning, these give the manuscript an overall colloquial tone, rather than a precise, scientific and authoritative tone. I am providing here a few examples, but there are multiple instances of such phrases on nearly every page:
- For nusinersen to be approved, it had to show effectiveness in several studies. - It has been corrected.
“In order to demonstrate the effectiveness of nusinersen, a lot of research was necessary. For nusinersen to be approved, it had to show effectiveness in several studies ​[18,23]​.“
- Below we present the promising results of several clinical trials that have approved the use of nusinersen. - It has been removed.
“Below we present the promising results of several clinical trials that have approved the use of nusinersen. The first clinical trial proving the efficacy of nursinersen, is the ENDEAR trial (ISIS=396443), started in 2014. “
- Apart from non-vector gene therapy, viral carriers have also found their place in clinical trials of ALS. Moreover, it turns out that they may represent a promising future. - It has been removed.
“Apart from non-vector gene therapy, viral carriers have also found their place in clinical trials of ALS. Moreover, it turns out that they may represent a promising future. So far, gene delivery to the central nervous system has been limited for most therapeutic molecules due to the existence of the blood-brain barrier. However, even this barrier can be overcome as demonstrated by Duque et al. through the use of AAV9 in animal studies [111]. “
- In the context of ALS gene therapy, the AAV9 serotype appears to be promising. - It has been corrected.
“As part of ALS gene therapy, serotype AAV9 seems promising. In the context of ALS gene therapy, the AAV9 serotype appears to be promising.”
- If their results prove to be positive, it could have a significant impact on the further development of gene therapy in ALS. - It has been removed.
“Currently, the safety, tolerability and efficacy of AAVrh10 intrathecal administration of anti-SOD1 microRNA to patients with SOD1 ALS mutations is being tested in the multi-center clinical trial APB-102 ​[132,133]​. If their results prove to be positive, it could have a significant impact on the further development of gene therapy in ALS.“
- It is worth mentioning that modulation of the SOD1 gene activity is not the end of the potential of viral vectors in ALS. - It has been removed.
“It is worth mentioning that modulation of the SOD1 gene activity is not the end of the potential of viral vectors in ALS. Additionally, An alternative approach could be the use of AAV vectors for the therapeutic delivery of the human gene DOK7. This is due to the fact that its expression causes the activation of muscle-specific kinase MuSK, which in turn inhibits the degradation of motor nerve endings at neuromuscular junctions.”

This manuscript is a resubmission of an earlier submission. The following is a list of the peer review reports and author responses from that submission.
Round 1
Reviewer 1 Report
The purpose of this review is to summarize the current state of gene therapy approaches for motor neuron diseases. There have been numerous review articles on this topic that have been published recently and this review does not really add any novel insights into this topic.
The description of the genetics of SMA is inaccurate. SMA is caused by the loss of SMN1 and retention of SMN2. The loss of both SMN1 and SMN2 is incompatible with life. This essential point is not clearly conveyed.
Figure 1 is misleading in that it implies that SMN2 pre-mRNA is somehow converted into full-length SMN1 mRNA with a frequency of 10-15%. This is clearly not the case but this figure does not accurately convey SMN2 alternative splicing.
The authors describe preclinical therapeutics testing of gene therapy approaches in different models for ALS, mainly cell line and mouse studies. Unfortunately, a similar type of preclinical development for approved FDA therapeutics.
Author Response
Response to Reviewer 1 Comments:
Dear Sir or Madam, thank you very much for the review our manuscript entitled: „ Gene therapy in the treatment of motor neuron diseases”
In response to your comment, we would like to thank you for appreciating our manuscript.
Comment 1
The description of the genetics of SMA is inaccurate. SMA is caused by the loss of SMN1 and retention of SMN2. The loss of both SMN1 and SMN2 is incompatible with life. This essential point is not clearly conveyed.
Revision and my comment
We have clarified the information on the pathogenesis of SMA, page 3; lines 101,102
Comment 2
Figure 1 is misleading in that it implies that SMN2 pre-mRNA is somehow converted into full-length SMN1 mRNA with a frequency of 10-15%. This is clearly not the case but this figure does not accurately convey SMN2 alternative splicing.
Revision and my comment
Figure 1 has been corrected to clearly show the difference in SMN1 and SMN2 splicing
Comment 3
The authors describe preclinical therapeutics testing of gene therapy approaches in different models for ALS, mainly cell line and mouse studies. Unfortunately, a similar type of preclinical development for approved FDA therapeutics.
Revision and my comment
Thank you very much for your comment, but we do not know how we improve our review. Could you explain us and help? We will be grateful for your help and indications.
We do honestly hope that it will satisfy you and improve the quality of our work. Once again, we are very grateful for your review and remain open if you have any other remarks or suggestions that will make our work merit publication in „International Journal of Molecular Science”.
Reviewer 2 Report
This is a review article with summerizing treatment nowadays and suggesting the trends in the near future. The ALS treatment could furter develope on the results of SMA treatments. I have some comments:
1.All 3 ways of SMA treatments are described, but the target SMA population is not enough defined - e.g. SMA 2, or 3. There are missing other important clinical scale (only Hammersmith scales - HINE)- RULM, walking distance
2.SMA classification is presented in one collumn -it could be more effective to present it in a table.
3.Reported clinical trials - Endear,etc - not reported classification and severity of SMA.
4. Only 3 figures for such complicated situation.
5.136 references - most of them recent ones.
6. Some abbreviations should be searching for longer time.
Author Response
Response to Reviewer 2 Comments:
Dear Sir or Madam, thank you very much for the review our manuscript entitled: „ Gene therapy in the treatment of motor neuron diseases”
In response to your comment, we would like to thank you for appreciating our manuscript.
Comment 1
All 3 ways of SMA treatments are described, but the target SMA population is not enough defined - e.g. SMA 2, or 3. There are missing other important clinical scale (only Hammersmith scales - HINE)- RULM, walking distance
Revision and my comment
We have added information on other scales for assessing motor development, pages 4-5, lines 192-221
Comment 2
SMA classification is presented in one collumn -it could be more effective to present it in a table.
Revision and my comment
We have added a table with the SMA classification
Comment 3
Reported clinical trials - Endear,etc - not reported classification and severity of SMA.
Revision and my comment
With each clinical trial description, the severity of SMA is described by specifying the type of disease or by specifying the number of copies of the SMN2 gene.
Comment 4
Only 3 figures for such complicated situation.
Revision and my comment
???
Comment 5
136 references - most of them recent ones.
Revision and my comment
Thank you very much for your comments.
Comment 6
Some abbreviations should be searching for longer time.
Revision and my comment
We corrected.
We do honestly hope that it will satisfy you and improve the quality of our work. Once again, we are very grateful for your review and remain open if you have any other remarks or suggestions that will make our work merit publication in „International Journal of Molecular Science”.
Round 2
Reviewer 1 Report
The potential impact of this manuscript is minimal given that there are many review articles written by established investigators in the field of motor neuron disease therapeutics. In other words, what value does this review add to the field and does it provide any novel insights that have not already been discussed in other publications?
The revised figure (Figure 1) showing the difference between SMN1 and SMN2, with respect to alternative splicing, is still incorrect.